# FrameMind: Frame-Interleaved Chain-of-Thought for Video Reasoning via Reinforcement Learning

## Abstract

Current video understanding models rely on fixed frame sampling strategies, processing predetermined visual inputs regardless of the specific reasoning requirements of each question. This static approach limits their ability to adaptively gather visual evidence, leading to suboptimal performance on tasks requiring either broad temporal coverage or fine-grained spatial detail. In this paper, we introduce **FrameMind**, a novel end-to-end framework trained with reinforcement learning that enables models to dynamically request visual information during reasoning through Frame-Interleaved Chain-of-Thought (FiCOT). Unlike traditional approaches, FrameMind operates in multiple turns where the model alternates between textual reasoning and active visual perception, using tools to extract targeted frames or video clips based on identified knowledge gaps. To train effective dynamic sampling policies, we propose Dynamic Resolution Frame Sampling (DRFS), which exposes models to diverse temporal–spatial trade-offs during learning, and DRFS-GRPO, a group-relative policy optimization algorithm that learns from outcome-based rewards without requiring frame-level annotations. Extensive experiments on challenging benchmarks like MLVU and VideoMME demonstrate that our method significantly outperforms existing models, advancing the state of the art in flexible and efficient video understanding.

## 1 Introduction

Multimodal Large Language Models (MLLMs) have demonstrated remarkable capabilities in static image understanding, achieving human-level performance on complex visual reasoning tasks(Alayrac et al., 2022; Liu et al., 2023; Li et al., 2023a; Yin et al., 2023). However, extending these successes to video understanding remains challenging due to the temporal complexity and computational constraints inherent in processing sequential visual content(Tang et al., 2023; Bertasius et al., 2021; Arnab et al., 2021; Tong et al., 2022). Videos require models to reason across time, tracking objects, events, and causal relationships while managing the trade-off between temporal coverage and spatial resolution under limited computational budgets(Lei et al., 2018; Xiao et al., 2021b; Yi et al., 2020; Feichtenhofer et al., 2019; Tang et al., 2025).

Most existing video MLLMs address this challenge by sampling a fixed set of frames before processing(Li et al., 2023c; Zhang et al., 2024b; Chen et al., 2024b; Zhang et al., 2024a; Ye et al., 2025; Shu et al., 2025), committing to a single sampling strategy regardless of the question's specific requirements (Hugging Face, 2024; Zhang et al., 2024c; Qian et al., 2024). This approach creates a fundamental disconnect: when analyzing a movie to identify "when does Tom catch Jerry," a model needs broad temporal coverage to scan the entire sequence, but when determining "what color is the laptop the woman is holding," it requires high spatial resolution of specific frames, which is demonstrated in Figure 1. Current approaches typically cannot adapt their perceptual strategy during reasoning, often resulting in either insufficient temporal scope for sequential questions or inadequate spatial detail for localized analysis.

To overcome these limitations, our key insight is that visual perception should be treated as an active, dynamic process rather than a static preprocessing step. Drawing inspiration from tool-augmented reasoning frameworks, we introduce Frame-Interleaved Chain-of-Thought (FiCOT), where models

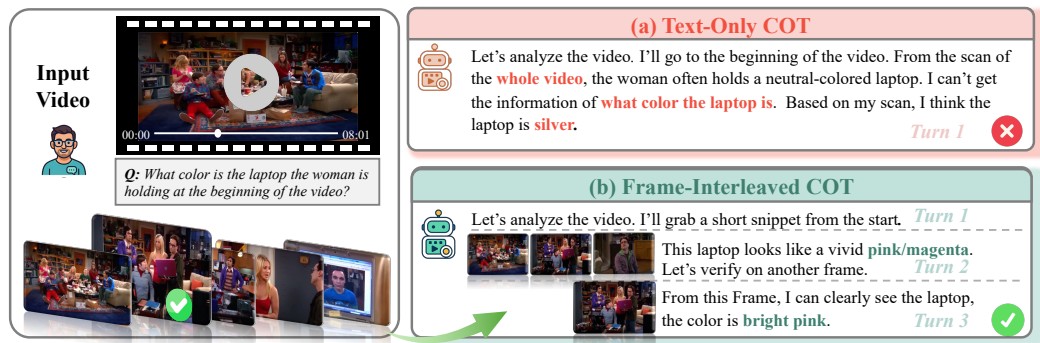

Figure 1: Comparison of a static, text-only CoT with our dynamic Frame-Interleaved CoT (FiCOT). (a) The conventional approach relies on a single, fixed scan of the video, resulting in insufficient spatial detail and an incorrect guess *silver*. (b) FiCOT actively identifies its knowledge gap and uses its toolbox to retrieve a high-resolution snippet and specific frames, leading to a grounded and correct answer *bright pink*.

alternate between textual reasoning and targeted visual evidence gathering. Instead of processing predetermined frames, the model can pause its reasoning, identify knowledge gaps, and actively request specific visual information from the video.

We realize this approach in FrameMind, an end-to-end agentic framework where the model operates through multiple reasoning turns, alternating between textual chain-of-thought and active visual tool calls. At each turn, the model can invoke tools to extract high-resolution frames at specific timestamps or sample frame sequences from targeted intervals, seamlessly integrating this evidence into its reasoning trajectory. Our key technical innovation is Dynamic Resolution Frame Sampling (DRFS), which trains the model across a resolution ladder spanning from low-resolution temporal scanning to high-resolution spatial focus, enabling adaptive perception policies. To train this complex decision-making process, we develop DRFS-GRPO, a group-relative policy optimization algorithm that learns effective sampling strategies from trajectory-level rewards, eliminating the need for expensive frame-level supervision(OpenAI et al., 2024; DeepSeek-AI et al., 2025).

We validate our approach on a diverse suite of challenging video understanding benchmarks, including MVBench, MLVU, and VideoMME. Our results demonstrate that FrameMind significantly outperforms existing open-source models and achieves performance competitive with top proprietary systems. Our main contributions are:

- We introduce **FiCOT**, a reasoning paradigm enabling dynamic visual evidence gathering during inference.
- We propose **DRFS**, a training methodology for learning adaptive sampling policies.
- We develop **DRFS-GRPO**, an efficient reinforcement learning algorithm for training complex perception-reasoning policies from sparse rewards.

## 2 RELATED WORKS

### 2.1 REASONING IN MULTIMODAL LLMS

RL-based post-training has been shown to enhance the reasoning abilities of large (multi)modal models through outcome-driven rewards and group-relative optimization (OpenAI et al., 2024; DeepSeek-AI et al., 2025). A substantial body of work extends this paradigm to multimodal settings by injecting perceptual evidence into the chain of thought: image-side studies expose intermediate spatial cues via visualization or editing (Li et al., 2025), and video-side investigations pursue reinforced video reasoning with multi-task evaluations and trajectory auditing (Feng et al., 2025). Beyond final-answer supervision, structured signals—such as format validity, tool-call correctness, and spatio-temporal alignment—are incorporated to alleviate supervision sparsity and spurious correlations (Zheng et al., 2025; Feng et al., 2025); stability is typically promoted via KL regularization

and preference/trajectory auditing (OpenAI et al., 2024; DeepSeek-AI et al., 2025). Notwithstanding these advances, perceptual granularity is often fixed, and attempts to scale RL to hour-level videos reveal unresolved choices concerning the allocation of visual bandwidth under tight computational budgets (Chen et al., 2025).

## 2.2 VIDEO UNDERSTANDING WITH MLLMS

Video understanding fundamentally requires balancing temporal coverage and spatial fidelity across mixed-length inputs. Earlier research emphasizes object-centric cues and iterative temporal selection/answering for long-form VideoQA (Gao et al., 2023a; Yu et al., 2023); complementary efforts reduce cost by compressing or sparsifying visual inputs—e.g., mapping each frame to a few tokens or employing streaming memory for very long sequences (Li et al., 2023c; Zhang et al., 2024a)—and by long-context finetuning that scales models to thousands of frames (Zhang et al., 2024b; Chen et al., 2024b). More recent systems couple adaptive scheduling with compression to unify short and long videos (Shen et al., 2024; Ye et al., 2025), while extra-long modeling advances the temporal horizon to hour-scale inputs (Shu et al., 2025). Benchmarking practice increasingly reports task accuracy alongside alignment and budget metrics to facilitate equal-budget comparisons, and RL-oriented studies explicitly target long-video regimes (Fu et al., 2025; Chen et al., 2025). Despite this progress, many pipelines still operate with static FPS/resolution presets or heuristic schedules, leaving open a systematic scheme for coordinating resolution and frame counts across heterogeneous lengths.

## 2.3 TOOL-AUGMENTED MLLMS

Equipping models with external tools extends capabilities beyond pure sequence modeling and establishes plan–act–reflect routines with auditable traces (Shen, 2024; Gao et al., 2023b; Schick et al., 2023). Within the image domain, callable visual operators—zoom/crop, detection/segmentation, sketch/edit—have been intertwined with reasoning: *DeepEyes*(Zheng et al., 2025) incentivizes visual tool use via reinforcement learning, *OpenThinkIMG*(Su et al., 2025) provides an end-to-end visual-tool RL framework, and *MVoT*(Li et al., 2025) treats visualization as an intermediate reasoning form (Zheng et al., 2025; Su et al., 2025; Li et al., 2025; Su, 2025). For videos, temporal grounding and indexing (e.g., manga-style frame/segment numbering) offer scaffolding for fine-grained evidence retrieval (Wu et al., 2025). Nevertheless, an effective mechanism by which models autonomously determine *when*, *what*, and *how* to invoke perception tools while reasoning over frames—under compute constraints and across mixed video lengths—remains insufficiently addressed.

# 3 METHOD

## 3.1 FRAMEWORK OVERVIEW

We propose FrameMind, a multimodal language model that performs video reasoning through an iterative perception-reasoning loop. Unlike conventional approaches that process a fixed set of pre-sampled frames, FrameMind acts as a dynamic agent, actively requesting visual information during the reasoning process based on its current understanding and identified knowledge gaps.

**Problem Formulation.** Given a video $V$ and question $Q$, conventional video MLLMs sample a fixed set of frames $F = \{f_1, f_2, ..., f_n\}$ and generate an answer $A$ through a single forward pass: $A = M(F, Q)$.

This static approach can be inefficient and prone to missing crucial details. In contrast, FrameMind operates through multiple turns, where each turn $k$ involves generating reasoning text $T_k$ and tool calls $C_k$, executing those calls to obtain new visual evidence $E_k$, and progressively refining its understanding.

**Core Components.** FrameMind consists of three key technical contributions: (1) Frame-Interleaved Chain-of-Thought (FiCOT), an iterative reasoning paradigm that interleaves textual reasoning with visual perception; (2) Dynamic Resolution Frame Sampling (DRFS), a training methodology that enables the model to reason effectively across different temporal-spatial trade-offs; and (3) DRFS-GRPO, a group-relative policy optimization algorithm for end-to-end training.

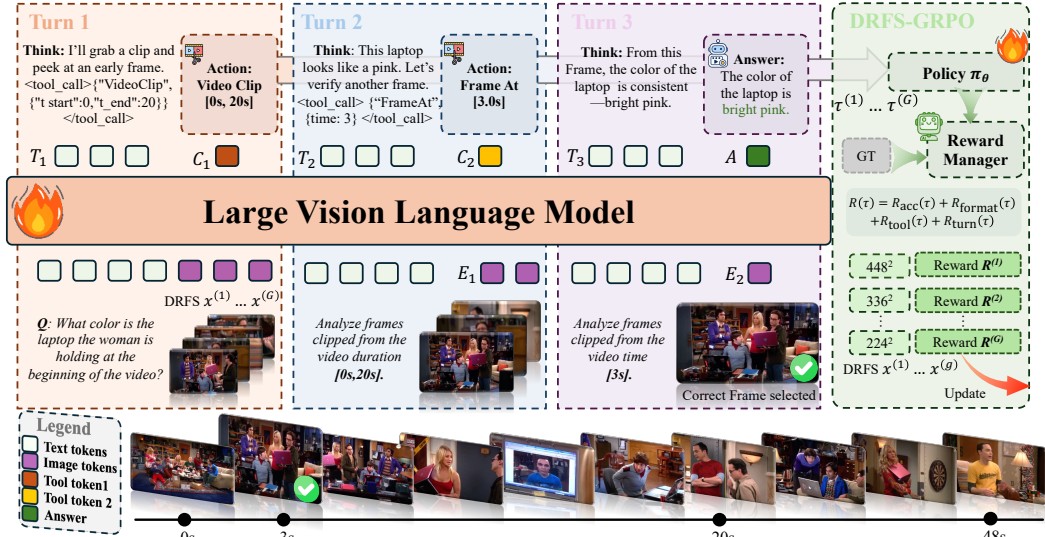

Figure 2: Overall framework of **FrameMind**, illustrating the iterative perception-reasoning loop. The agent first thinks, then acts (calls tools) to gather visual evidence, and updates its understanding to inform the next cycle.

## 3.2 FRAME-INTERLEAVED CHAIN-OF-THOUGHT (FICOT)

At the heart of our agent is the Frame-Interleaved Chain-of-Thought (FiCOT) process, which reformulates video reasoning from a single-step generation into a multi-turn dialogue between reasoning and perception. This grounds the model's internal monologue in concrete visual evidence. The complete reasoning trajectory is defined as $\tau = \{T_1, C_1, E_1, T_2, C_2, E_2, ..., T_n, A\}$, capturing each step of thought $T_k$, action $C_k$, observation $E_k$, and final answer $A$.

**Turn-based Execution.** Each turn k follows a three-stage process:

*Stage 1: Generation.* At the start of each turn $k$, the policy, denoted as $\pi_\theta$, receives two inputs: the complete textual history $H_{k-1} = \{Q, T_1, \ldots, T_{k-1}\}$, containing the initial question $Q$ and all prior reasoning steps, and the visual evidence $E_{k-1}$ gathered from the previous turn. Based on this information, the policy generates the next textual thought $T_k$ and a corresponding set of tool calls $C_k$. This is formally expressed as:

$$(T_k, C_k) \sim \pi_\theta(\cdot \mid H_{k-1}, E_{k-1}) \tag{1}$$

For the initial turn ($k = 1$), the model starts with a set of uniformly sampled frames from the video, denoted as $E_0$.

*Stage 2: Tool Execution.* When the model's reasoning output contains a special `<tool_call>` tag, a parser extracts the enclosed action instruction, $C_k$. This instruction is then mapped to one of the two available tools for execution:

- `FrameAt(t)`: To zoom in on a specific moment `t` with a resolution (448×448) frame.
- `VideoClip(t_start, t_end)`: To scan a time interval from `t_start` to `t_end` with a sequence of 8-20 frames.

The visual information returned by the executed tool is then used to populate the evidence set $E_k$, which serves as the visual input for the next reasoning turn $k + 1$.

*Stage 3: State Update.* The newly acquired visual content is organized into a timestamped evidence set $E_k = \{(f_i, t_i)\}$ and fed back to the model for the next turn.

**Termination.** The process terminates when: (1) the model generates a response containing the special token `<answer></answer>`, or (2) the maximum number of turns (3 in our implementation) is reached.

---

**Algorithm 1** DRFS-GRPO Training Step

---

**Require:** Batch $\{(V_i, q_i)\}_{i=1}^B$, group size $G$
1: **for** $i = 1$ to $B$ **do**
2:     **for** $g = 1$ to $G$ **do**                                     ▷ Build DRFS ladder
3:         $r \leftarrow \frac{g-1}{G-1}$; generate $x_i^{(g)}$ using resolution ladder
4:     **end for**
5:     **for** $g = 1$ to $G$ **do**                                     ▷ Execute & evaluate
6:         $\tau_i^{(g)} \sim \pi_\theta(\cdot \mid x_i^{(g)}, q_i)$; compute $R_i^{(g)}$
7:     **end for**
8:     $\bar{R}_i \leftarrow \frac{1}{G} \sum_g R_i^{(g)}$;  $A_i^{(g)} \leftarrow R_i^{(g)} - \bar{R}_i$              ▷ Group-relative advantage
9:     Update policy using PPO objective with $A_i^{(g)}$
10: **end for**

---

**Training Efficiency.** Unlike approaches requiring expensive frame-level annotations, FiCOT only requires video-question-answer triplets for outcome-based reward signals, making it scalable to large datasets.

### 3.3 DYNAMIC RESOLUTION FRAME SAMPLING (DRFS)

While FiCOT provides the agentic loop, its effectiveness depends entirely on the quality of the visual evidence it receives. A fundamental challenge here is the trade-off between temporal coverage and spatial detail. Locating a brief event in a long video requires scanning many low-resolution frames, while understanding a complex, rapid action requires focusing on a few high-resolution frames.

To equip our model with the versatility to handle both scenarios, we introduce Dynamic Resolution Frame Sampling (DRFS). Instead of training on a single, fixed sampling strategy, DRFS exposes the model to a whole spectrum of possibilities during each training step.

**Resolution Ladder Construction.** For any video during training, DRFS generates $G$ parallel visual inputs spanning a spectrum of sampling strategies. Let $(N_L, H_L, W_L)$ and $(N_H, H_H, W_H)$ represent the low-resolution (many frames, small size) and high-resolution (few frames, large size) endpoints respectively. For each group member $g \in \{1, 2, \ldots, G\}$, we set an interpolation weight $r \in [0, 1]$ that increases linearly with $g$, i.e., $r = \frac{g-1}{G-1}$ (Algorithm 1, line 3). We then compute:

$$N_g = (1 - r)N_L + rN_H \tag{2}$$
$$(H_g, W_g) = (1 - r)(H_L, W_L) + r(H_H, W_H) \tag{3}$$

This creates a "resolution ladder" where $g = 1$ corresponds to temporal scanning and $g = G$ corresponds to spatial focus.

**Frame Sampling and Resizing.** For each configuration $(N_g, H_g, W_g)$, we sample $N_g$ frames uniformly, resize each to $H_g \times W_g$, and stack them into the input tensor $x^{(g)}$.

**Training Rationale.** By forcing the model to reason across this entire spectrum, DRFS enables the policy to learn which sampling strategy is most effective for different video types and questions, making it robust to diverse evaluation scenarios.

### 3.4 REINFORCEMENT LEARNING WITH DRFS-GRPO

To learn a policy $\pi_\theta$ that can effectively utilize the diverse visual inputs from DRFS, we employ reinforcement learning. However, standard RL algorithms are not designed to handle the group of parallel rollouts generated by our DRFS methodology. We therefore propose DRFS-GRPO (Group Relative Policy Optimization), an algorithm specifically designed to leverage this group structure for more effective and efficient policy learning.

**Group-Relative Advantage Computation.** To teach the model which initial sampling strategy is most effective, our process begins with the group of $G$ parallel visual inputs, $\{x^{(1)}, \ldots, x^{(G)}\}$, generated by the DRFS resolution ladder. For each of these distinct inputs, the policy executes a full reasoning trajectory $\tau_i^{(g)}$, resulting in $G$ parallel rollouts for the same video-question pair. After

obtaining the final reward $R_i^{(g)}$ for each rollout, we compare their performance. The core idea of DRFS-GRPO shown in Algorithm 1 is to judge each strategy's outcome relative to its peers by calculating a group-average reward $\bar{R}_i$. The advantage $A_i^{(g)}$ for each rollout is then its performance relative to this group average.

**Policy Optimization.** The policy uses this group-relative advantage in a PPO-style update. Actions from trajectories with a positive advantage ($A_i^{(g)} > 0$), which performed better than the group average, are reinforced, while those with a negative advantage are suppressed. This direct comparison of parallel outcomes efficiently teaches the policy to select the best sampling strategies for different videos and questions.

$$\mathcal{J}(\theta) = \mathbb{E}_{(V,q)\sim\mathcal{D}}$$

$$\left[ \frac{1}{G} \sum_{g=1}^{G} \sum_{t=1}^{|\tau^{(g)}|} \min\left( r_t^{(g)}(\theta) A_i^{(g)}, \text{clip}(r_t^{(g)}(\theta), 1-\varepsilon, 1+\varepsilon) A_i^{(g)} \right) - \beta D_{\text{KL}}(\pi_\theta \| \pi_{\text{ref}}) \right] \quad (4)$$

where $r_t^{(g)}(\theta) = \frac{\pi_\theta(a_t|s_t)}{\pi_{\theta_{\text{old}}}(a_t|s_t)}$ is the probability ratio for the action at timestep $t$ in rollout $g$, $A_i^{(g)}$ is the group-relative advantage for the entire trajectory, and the final term is a KL divergence regularizer for training stability.

### 3.5 REWARD DESIGN

The DRFS-GRPO algorithm trains the policy to maximize a cumulative reward. The design of this reward function is crucial, as it defines what a "good" trajectory looks like. We designed a function that guides the agent toward not just correct answers but also efficient and structurally sound reasoning.

**Core Objective.** The primary signal is task success, given by the Accuracy Reward ($+1$ for a correct answer) and a Format Penalty ($-1$ for improper structure) to ensure valid output.

$$R_{\text{acc}}(\tau) = \begin{cases} 1 & \text{if final answer is correct} \\ 0 & \text{otherwise} \end{cases} \quad (5)$$

$$R_{\text{format}}(\tau) = \begin{cases} 0 & \text{if format is valid} \\ -1 & \text{otherwise} \end{cases} \quad (6)$$

**Behavioral Shaping.** To encourage intelligent behavior, we add two shaping rewards. The **Tool Usage Incentive** ($R_{tool}$) encourages using the available tools effectively. It is based on a raw tool score, $s_{\text{tool}}(\tau)$, which grants a score of 1.0 for using one unique tool type and a synergy bonus for a total score of 1.2 if both are used. This score is then gated by the final answer's correctness, with a small base reward granted for exploration and a larger reward for tool use that leads to a correct answer. The **Efficiency Bonus** rewards the agent for finding the answer in a reasonable number of steps (2 or 3 turns).

$$R_{\text{tool}}(\tau) = s_{\text{tool}}(\tau) \times \left( 0.2 + 0.8 \cdot R_{\text{acc}}(\tau) \right) \quad (7)$$

$$R_{\text{turn}}(\tau) = 0.5 \cdot \mathbb{I}\left[ 1 < |\text{turns}(\tau)| \leq 3 \right] \quad (8)$$

**Total Reward.** The final reward is a sum of these components, creating a balanced objective for the agent to pursue :

$$R(\tau) = R_{\text{acc}}(\tau) + R_{\text{format}}(\tau) + R_{\text{tool}}(\tau) + R_{\text{turn}}(\tau) \quad (9)$$

## 4 EXPERIMENTS

### 4.1 EXPERIMENTAL DETAILS

We build on Qwen2.5-VL-7B(Bai et al., 2025) as the base policy and train with the EasyR1 reinforcement learning framework. We implement a multi-turn dialogue protocol capped at three turns, keeping recent turns as in-context memory. For perception, we adopt Dynamic-Resolution Frame Sampling (DRFS): each trajectory samples 32–64 frames with linear interpolation of spatial resolution from 224×224 (low-res) to 448×448 (high-res). For targeted inspection, the `VideoClip` tool

uniformly extracts 8–20 frames over a specified time span and resizes all frames to 448×448. Unless noted, DRFS inputs and tool clips are fused within the same turn under the three-turn protocol. All experiments run on a single server with 8×NVIDIA A100-80GB GPUs.

## 4.2 TRAINING DATA FOR FRAMEMIND

**Data for Agentic Reasoning.** We curate ∼7.6K video–QA instances to elicit active visual probing rather than single-pass decoding. The mix includes fine-grained perception (Perception-Test (Pătrăucean et al., 2023)), everyday narratives (LLaVA-Video-178K subset (Zhang et al., 2024c)), spatio-temporal relations (STAR (Wu et al., 2024)), physical causality and counterfactuals (CLEVRER (Yi et al., 2019)), and event-centric explanations (NeXT-QA (Xiao et al., 2021a)). These tasks require locating decisive moments/objects, prompting the agent to call `VideoClip` for coarse temporal localization and `FrameAt` for high-resolution inspection.

**Data for DRFS Training.** To teach resolution–coverage trade-offs, we pair long-form LongVideo-Reason and rationale-rich VideoEspresso (favoring low-res "global scans") with short/mid clips from NeXT-QA and STAR (rewarding high-res "precise focus") (Chen et al., 2025; Han et al., 2025; Xiao et al., 2021a; Wu et al., 2024). This mix is essential for training the DRFS ladder to balance temporal coverage and spatial detail.

**Dataset Curation and Preprocessing.** Our final ∼7.6K instances support generating full reasoning trajectories $\tau = \{\mathcal{T}_1, \mathcal{C}_1, \mathcal{V}_1, \ldots, \mathcal{A}_n\}$. While the data mainly provides (video, question, final_answer) for the outcome-driven accuracy reward $R_{acc}$, reinforcement learning operates on the complete agent-generated trajectories. Preprocessing includes normalizing video resolutions for a consistent DRFS baseline, instantiating candidate sampling configurations, and applying decontamination and batch reweighting to ensure quality and domain balance (Zhang et al., 2024c; Wu et al., 2024; Yi et al., 2019; Pătrăucean et al., 2023; Xiao et al., 2021a; Chen et al., 2025; Han et al., 2025).

## 4.3 BENCHMARKS

To comprehensively evaluate our proposed DRFS strategy, we selected three benchmarks that collectively span a wide spectrum of video durations and reasoning tasks. A more detailed description of each benchmark is provided in Appendix C.

**Video-MME.** (Fu et al., 2025) is a comprehensive benchmark for multi-modal understanding. It assesses both foundational perception and higher-order cognitive reasoning across a wide spectrum of video durations, from short clips to hour-long videos.

**MLVU.** (Zhou et al., 2025) is a multi-task benchmark specifically targeting the challenges of long-video understanding. Its tasks require both global comprehension of long narratives and fine-grained temporal localization, featuring "needle-in-a-haystack" questions that test a model's ability to recall specific details.

**MVBench.** (Li et al., 2024b) is designed to evaluate fine-grained temporal perception in short video clips. It consists of 20 challenging, multiple-choice tasks, such as identifying action sequences and state changes, to pinpoint a model's ability to process moment-to-moment details.

## 4.4 BASELINES

We compare FrameMind against a comprehensive set of baselines, which we group into three categories based on their training methodology. For a detailed description of each model, please refer to Appendix D.

**Standard Video MLLMs.** We compare against a range of models trained with supervised fine-tuning. These include general-purpose models like Video-LLaVA (Lin et al., 2023) and VideoChat2 (Li et al., 2023b), as well as models specialized for long-context or efficient processing like LongVA (Zhang et al., 2024b) and LLaMA-VID (Li et al., 2023d). The full list includes Chat-UniVi (Jin et al., 2024), ShareGPT4Video (Chen et al., 2024a), LLaVA-NeXT-Video (Li et al., 2024a), Video-LLaMA2 (Cheng et al., 2024), Video-CCAM (Fei et al., 2024), and Video-XL (Shu et al., 2025).

**Reinforcement Learning-Based Models.** To situate our work, we compare against Video-R1 (Feng et al., 2025), another recent model that uses reinforcement learning to enhance reasoning. We use Qwen2.5-VL-7B (Bai et al., 2025) as the base model for our RL training.

| Models | Size | Frames | MVBench | MLVU | VideoMME (w/o.sub) | |
|---|---|---|---|---|---|---|
| | | | | Test | Overall | Long |
| *Duration* | – | – | 5∼35 s | 3∼120 m | 1∼60 m | 30∼60 m |
| **Proprietary Models** | | | | | | |
| GPT4-V (OpenAI, 2023) | – | 1fps | 43.5 | – | 60.7 | 56.9 |
| GPT4-o (OpenAI, 2024) | – | 0.5fps | – | **54.9** | **77.2** | **72.1** |
| Gemini 1.5 Pro(Gemini Team, 2024) | – | 0.5fps | – | – | 75.0 | 67.4 |
| **Open-Source Video MLLMs** | | | | | | |
| Video-LLaVA (Lin et al., 2023) | 7B | 8 | 41.0 | 30.7 | 40.4 | 38.1 |
| LLaMA-VID (Li et al., 2023d) | 7B | 1fps | 41.9 | 33.2 | – | – |
| VideoChat2 (Li et al., 2023b) | 7B | 16 | 51.1 | 35.1 | 39.5 | 33.2 |
| Chat-UniVi (Jin et al., 2024) | 7B | 64 | – | – | 40.6 | 35.8 |
| ShareGPT4Video (Chen et al., 2024a) | 8B | 16 | 51.2 | 46.4 | 43.6 | 37.9 |
| LLaVA-NeXT-Video (Li et al., 2024a) | 7B | 32 | 33.7 | – | 46.5 | – |
| VideoLLaMA2 (Cheng et al., 2024) | 8×7B | 32 | 54.6 | 45.6 | 46.6 | 43.8 |
| LongVA (Zhang et al., 2024b) | 7B | 128 | 52.3 | 41.1 | 52.6 | 46.2 |
| Video-CCAM (Fei et al., 2024) | 9B | 16/96 | **64.6** | 42.9 | 50.3 | 39.6 |
| Video-XL(Shu et al., 2025) | 7B | 128 | 55.3 | 45.6 | 52.3 | 48.9 |
| Qwen2.5-VL-7B(Bai et al., 2025) | 7B | 32 | 62.6 | 41.6 | 53.6 | 44.7 |
| Video-R1(Feng et al., 2025) | 7B | 32 | 63.9 | 45.4 | 59.3 | 50.2 |
| **FrameMind (Ours)** | 7B | 32 | 64.2 | **48.6** | **60.9** | **57.5** |

Table 1: Performance comparison on key video understanding benchmarks. All scores are reported as accuracy (%). The best and second-best performance among open-source models in each column is marked in **bold** and underlined respectively.

**Proprietary Models.** We also benchmark against powerful, closed-source models, including OpenAI's GPT-4V and GPT-4o (OpenAI, 2024), and Google's Gemini 1.5 Pro (Gemini Team, 2024), to contextualize our performance against the state-of-the-art.

## 4.5 MAIN RESULTS

To ensure a standardized assessment, we evaluate FrameMind on benchmarks featuring objective, multiple-choice questions, reporting high accuracy in Table 1. We compare against two groups: leading proprietary systems and open-source models. The results show that FrameMind consistently delivers competitive or superior performance, establishing a new state-of-the-art among open-source peers under comparable configurations.

FrameMind's strong performance across videos of diverse lengths is a direct result of its learned ability to *thinking with frames* Through our proposed FiCOT process and DRFS mechanism, the model learns an adaptive perception policy. On short-video benchmarks like MVBench, which require high spatial fidelity, the model learns to use its toolbox to perform a high-resolution "precise focus" on specific frames to capture fine-grained details. Conversely, on long-form benchmarks like MLVU, it learns to deploy a low-resolution "global scan" to efficiently explore broad temporal regions for contextual understanding. This learned ability to intelligently trade spatial detail for temporal coverage is what drives its robust performance.

Notably, on the short-form **MVBench**, FrameMind achieves a secondary accuracy of 64.2%, surpassing all other open-source models and highlighting its superior temporal perception. In the long-video domain, our model achieves 48.6% accuracy on the test of **MLVU**, outperforming specialized long-video models like LongVA while using up to 4x fewer frames. On the comprehensive **VideoMME** benchmark, FrameMind not only leads the open-source field with an overall accuracy of 60.9% but also surpasses the performance of GPT-4V. These results underscore the effectiveness of training a model to actively make perceptual decisions, leading to superior performance and efficiency compared to methods that rely on static visual inputs.

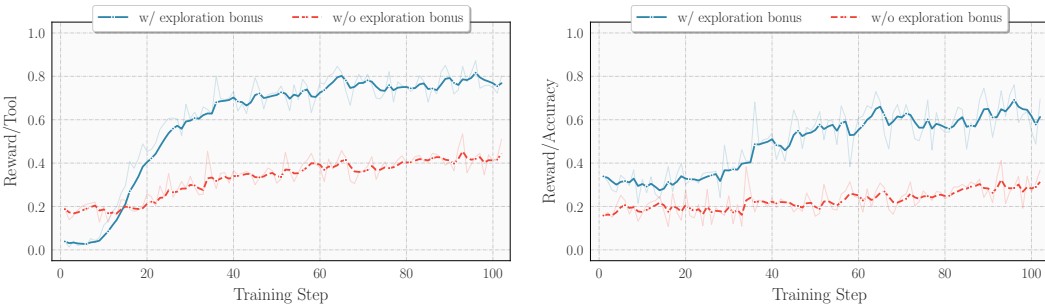

Figure 3: **Effect of the exploration bonus.** *(Left)* reward/tool and *(Right)* reward/accuracy over training steps. With the exploration bonus (blue), the curves take off earlier and converge to a higher plateau on both metrics; without it (red), learning is slower and plateaus lower.

## 4.6 ABLATION STUDY

In this section, We conduct a comprehensive ablation studies to evaluate the impact of key components of FrameMind on VideoMME.

**Analysis of the DRFS-GRPO Training.** To validate the effectiveness of our DRFS-GRPO training methodology, particularly the "resolution ladder", we conduct an ablation against a Standard GRPO baseline. This baseline uses the same group-relative optimizer but is trained on a fixed 32-frame sampling configuration, without the diverse inputs provided by the DRFS ladder. The results in Table 2 reveal a significant performance gap. Under identical evaluation budgets, DRFS consistently and significantly outperforms the GRPO baseline, delivering gains of **+6.9**

| Train Set | VideoMME (w/o sub.) | | | |
|---|---|---|---|---|
| | **Short** | **Medium** | **Long** | **Overall** |
| GRPO-32 | 58.0 | 54.5 | 49.5 | 54.0 |
| GRPO-48 | 58.5 | 55.0 | 50.0 | 54.5 |
| GRPO-64 | 60.0 | 57.5 | 51.5 | 56.3 |
| **DRFS-32** | **63.8** | **61.4** | **57.5** | **60.9** |
| **DRFS-48** | **65.5** | **64.1** | **60.0** | **63.2** |
| **DRFS-64** | **66.0** | **64.8** | **61.2** | **64.0** |

Table 2: VideoMME by duration. 32/48/64 denote frames at evaluation.

**(Overall) and +8.0 (Long)** points at just 32 frames. This result demonstrates that the DRFS ladder is crucial for teaching the model a robust and flexible policy; without being forced to reason across a spectrum of visual fidelities and having the most successful strategies amplified by the optimizer, the model fails to develop the adaptability needed for diverse video tasks. This enhanced robustness is especially evident on long videos, where DRFS maintains a much higher **Long-to-Overall accuracy ratio** (e.g., **0.944 vs. 0.917** at 32 frames), indicating a markedly smaller performance drop on challenging long-duration inputs.

**Necessity of the Exploration Bonus.** Our goal-gated reward grants a small, unconditional 20% bonus for tool use to encourage exploration. To validate this design choice, we test a variant with **Strict Gating**, where this exploration bonus is removed ($R_{\text{tool}} = s_{\text{tool}} \cdot R_{\text{acc}}$). In this setting, the tool reward is zero unless the final answer is correct. We observe that this model struggles to learn a robust tool-use policy because the reward signal becomes overly sparse. As illustrated in Figure 3, the model trained with the exploration bonus (blue line) shows a steady increase in both its tool-use reward and final task accuracy. In contrast, the model with Strict Gating (red line) fails to learn an effective tool-use policy, and its accuracy stagnates at a low level. This experiment confirms that the 20% exploration bonus is a critical component for mitigating reward sparsity and bootstrapping the learning process in a complex, tool-augmented environment.

## 5 CONCLUSION

In this work, we presented **FrameMind**, a framework designed to overcome the static perception limitations of current video models. Our approach, Frame-Interleaved Chain-of-Thought (FiCOT), enables a model to actively gather visual evidence during its reasoning process. Trained end-to-end with our DRFS-GRPO reinforcement learning algorithm, FrameMind learns to adapt its perceptual strategy, deciding whether to perform a broad temporal scan or a high-resolution focus. Experiments show our method achieves state-of-the-art results, representing a key step toward more flexible and efficient video understanding model that can decide not just what to think, but how to look.

## ETHICS STATEMENT

All authors adhere to the ICLR Code of Ethics. This work uses only publicly released research datasets cited in the paper; we do not collect new human-subjects data and do not process personally identifying information beyond what may exist in those datasets. We do not perform identity recognition or deploy the system; results are reported solely for research under equal-budget settings. Potential risks include dataset biases and misuse; we mitigate by reporting objective metrics and ablations, documenting limitations, and releasing only code/configs/metadata (not raw video). Compute and procedures are described in the *Experiments* section; we reduce environmental impact via moderate budgets, gradient checkpointing, and fixed seeds. Large language models were used only for writing polish (see Appendix A); all scientific claims, code, and experiments were designed and verified by the authors.

## REPRODUCIBILITY STATEMENT

Implementation details for FiCOT/DRFS and Algorithm 1 appear in Section 3; training hyperparameters are in Table E1; data composition and preprocessing in Section 4.2; benchmark protocols in Appendix C; baselines in Appendix D; additional analyses in Appendix E.4. Main results and ablations are in Table 1, Table 2, and Figure 3. After the review period ends, we will open-source the codebase, DRFS utilities, prompts, exact configs and seeds, and scripts to prepare all public datasets, enabling faithful reproduction of our numbers under the same budgets.

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

# FRAMEMIND: FRAME-INTERLEAVED CHAIN-OF-THOUGHT FOR VIDEO REASONING VIA REINFORCEMENT LEARNING

# APPENDIX

In this supplementary document, we provide additional details and experimental results to enhance understanding and insights into our method. This supplementary document is organized as follows:

## A  LLM USAGE

I have used large language models just to polish my paper writing.

## B  METHOD IMPLEMENTATION DETAILS

This section describes the methodology details of the proposed FrameMind framework in Section 3. We first illustrate the implementation details of the video toolbox and then demonstrate the training details.

### B.1  DETAILED REWARD FUNCTION SETUP

Our reward balances final task accuracy with well-structured, executable tool usage and concise multi-turn reasoning. The total episode reward is

$$R(\tau) = R_{\text{acc}} + R_{\text{format}} + R_{\text{tool}} + R_{\text{turn}}. \tag{10}$$

**Accuracy Reward ($R_{\text{acc}}$).**  Primary signal for task success:

$$R_{\text{acc}} = \begin{cases} 1, & \text{if the final answer is correct,} \\ 0, & \text{otherwise.} \end{cases} \tag{11}$$

**Format Enforcement ($R_{\text{format}}$).**  We require the whole response to be one or more closed `<think></think>` blocks followed by exactly one closed `<answer></answer>` block (and nothing after it). Let the format-validity indicator be

$$\mathbb{I}_{\text{fmt}} = \begin{cases} 0, & \text{all } \texttt{<think>} \text{ are closed and there is exactly one closed } \texttt{<answer>} \text{ at the end,} \\ -1, & \text{otherwise.} \end{cases} \tag{12}$$

Then the format term is `<think>...</think>...<answer>...</answer>`.

**Tool Usage Incentive ($R_{\text{tool}}$).**  A response from trajectory may contain one or more closed `<tool_call></tool_call>` blocks. Each must be (i) well-formed, (ii) regex-parsable into a normalized schema, and (iii) executable for *correct* frame/segment extraction.

**Turn Efficiency Bonus ($R_{\text{turn}}$).**  Let $T$ be the number of closed `<turn_sum></turn_sum>` blocks in the final response. Reward concise multi-turn reasoning:

$$R_{\text{turn}} = \lambda_{\text{turn}} \cdot \mathbb{1}[\, 2 \le T \le 3\,], \qquad \lambda_{\text{turn}} = 0.5. \tag{13}$$

**Rollout Turn Control (Loop Policy).**  Each turn produces `<think>···</think>` (and optional `<tool_call>···</tool_call>`); it either emits `<turn_sum>···</turn_sum>` to *continue*, or a terminal `<answer>···</answer>` to *stop*. The controller is:

1. Start at turn $k = 1$.

2. If the previous turn emitted a closed `<turn_sum>` and $k < 3$, proceed to turn $k+1$; else stop.

3. If any turn emits a closed `<answer>`, stop immediately.

4. Hard cap at 3 turns (i.e., if $k = 3$ finishes without `<answer>`, stop).

### B.2 Tool Implementation and Parsing

The FrameMind agent is equipped with a concise yet powerful toolbox designed for targeted visual evidence gathering. The implementation of these tools and the parsing mechanism are as follows:

### B.3 Tool Implementation and Parsing

The FrameMind agent is equipped with a concise yet powerful toolbox designed for targeted visual evidence gathering. The implementation of these tools and the parsing mechanism are as follows:

- **Tool Definitions**: The agent has access to two primary tools for visual perception:
  - `FrameAt(t)`: This tool is designed for high-detail spatial inspection. Given a specific timestamp `t` (in seconds), it extracts the single closest frame from the video and resizes it to a high resolution of $448 \times 448$ pixels. This is used when the model needs to "zoom in" on a precise moment.
  - `VideoClip(t_start, t_end)`: This tool is used for temporal scanning of a specific interval. Given a start time `t_start` and an end time `t_end`, it uniformly samples a sequence of 8 to 20 frames from within that duration. These frames are also resized to $448 \times 448$ to provide high-resolution context for the selected segment.
- **Parsing and Execution**: The model invokes these tools by generating a special `<tool_call>` tag within its reasoning output. A parser, implemented using regular expressions, then extracts the function call (e.g., `VideoClip(15.5, 20.0)`) from within this tag.
- **Error Handling**: If a tool call is malformed (e.g., incorrect syntax, invalid parameters like `t_end < t_start`, or a timestamp outside the video's duration), the tool execution engine returns a concise error message (e.g., `ERROR: Invalid timestamp. Video duration is 60s.`). This feedback is appended to the dialogue history, allowing the agent to recognize its mistake and attempt a corrected action in the subsequent turn.

## C Benchmark Details

We evaluate our model on three key video understanding benchmarks, each with a distinct focus.

**Video-MME.** (Fu et al., 2025) is a comprehensive benchmark for multi-modal video understanding that assesses both perceptual and cognitive abilities. It contains 900 videos totaling 254 hours, with lengths from 11 seconds to over an hour. The dataset spans six diverse visual domains (e.g., movies, vlogs, sports) and 30 subfields, accompanied by 2,700 multiple-choice questions. It is designed to test foundational perception tasks like action and attribute recognition, as well as higher-order cognition, including character relationship analysis, commonsense reasoning, and scene understanding. The benchmark holistically evaluates models by accepting inputs from various modalities, including video frames, subtitles, and audio.

**MLVU.** (Zhou et al., 2025) is a multi-task benchmark specifically targeting the challenges of long-video understanding. It comprises 1,730 videos, with an average duration of 15 minutes and some exceeding 2 hours, paired with 3,102 questions across nine distinct task categories. These tasks, such as Anomaly Recognition, Plot QA, and First-person QA, require a mix of global, long-range reasoning and fine-grained temporal localization. A notable feature is its "needle-in-a-haystack" questions, which test a model's ability to recall specific, localized details from an extensive video context. The benchmark includes both multiple-choice and open-ended generative questions, split into development and test sets.

**MVBench.** (Li et al., 2024b) is a benchmark designed to evaluate fine-grained temporal under-standing in short video clips. It consists of 20 challenging tasks that require detailed temporal perception and reasoning, such as identifying action sequences, recognizing state changes, and un-derstanding object interactions. The tasks are presented in a multiple-choice format and are designed to pinpoint specific model capabilities in processing continuous video streams. By focusing on short-form content, MVBench serves as a crucial test for a model's ability to capture precise, moment-to-moment details, complementing the long-range reasoning required by other benchmarks.

## D  BASELINES DETAILS

**Standard Video MLLMs.** This group comprises models primarily trained with supervised fine-tuning. Video-LLaVA (Lin et al., 2023) employs a two-stage curriculum, first pre-training on images and then fine-tuning on video instruction data. VideoChat2 (Li et al., 2023b) is a video-centric model that utilizes a spatiotemporal-aware visual encoder and is trained on diverse, high-quality video in-struction data. Chat-UniVi (Jin et al., 2024) learns a unified representation for images, video, and audio through a shared vector quantizer. ShareGPT4Video (Chen et al., 2024a) is a model trained on a large-scale, high-quality dataset of human-annotated video dialogues. LLaVA-NeXT-Video (Li et al., 2024a) improves upon the LLaVA architecture with higher-resolution visual encoders and enhanced visual instruction tuning. VideoLLaMA2 (Cheng et al., 2024) integrates spatiotempo-ral, audio, and speech modalities for a holistic understanding. For models targeting diverse video lengths, Video-CCAM (Fei et al., 2024) introduces a causal cross-attention mask to improve per-formance on both short and long videos. For efficiency and long-context processing, LLaMA-VID (Li et al., 2023d) generates a single context token to represent an entire video's content, while LongVA (Zhang et al., 2024b) and Video-XL (Shu et al., 2025) employ coarse-to-fine aggregation and attention-sparsification techniques to handle thousands of frames.

**Reinforcement Learning-Based Models.** This category includes models that use reinforcement learning (RL) to enhance reasoning. Video-R1 (Feng et al., 2025) is an end-to-end agentic reasoning framework trained with a novel group-relative policy optimization algorithm on a multi-task dataset. Our work extends this paradigm by training FrameMind to learn a dynamic perception policy, using Qwen2.5-VL-7B (Bai et al., 2025) as our base model.

**Proprietary Models.** To contextualize our performance, we also report scores from leading pro-prietary models. These include OpenAI's GPT-4V and GPT-4o (OpenAI, 2024), and Google's Gem-ini 1.5 Pro (Gemini Team, 2024), which is distinguished by its extremely large context window.

## E  EXPERIMENT DETAILS

### E.1  PROMPTING FORMAT

To facilitate the Frame-Interleaved Chain-of-Thought (FiCOT) process, we structure the input to the model as a multi-turn dialogue. At the beginning of each reasoning turn $k$, the model receives a formatted prompt that includes the entire history of the interaction. This includes the initial question, all previous textual thoughts, tool calls, and the visual evidence gathered.

The visual evidence $E_{k-1}$ from the previous turn, which is a set of timestamped frames $\{(f_i, t_i)\}$, is integrated directly into the context. The general template is as follows:

```
972   SYSTEM_PROMPT
973
974     {{ content | trim }} You are an expert video analysis assistant.
975
976     # Tools
977     You are provided with function signatures within <tool_call></
978         tool_call> XML tags:
        <tool_call>
979     {
980       "type": "function",
981       "function": {
982         "name": "FrameAt",
983         "parameters": {
984           "type": "object",
985           "properties": {
986             "time": {
987               "type": "number",
988             }
989           },
990           "required": ["time"]
991         }
        }
992     }
993     </tool_call>
        <tool_call>
994     {
995       "type": "function",
996       "function": {
997         "name": "VideoClip",
998         "parameters": {
999           "type": "object",
1000          "properties": {
              "t_start": { "type": "number", "Start time (s)" },
1001          "t_end": { "type": "number", "End time (s)" }
            },
1002        "required": ["t_start", "t_end"]
1003        }
1004      }
1005    }
        </tool_call>
1006
1007    # How to call a tool
1008    Return a json object with function name and arguments within <
1009        tool_call></tool_call> XML tags:
1010    <tool_call>
        { "name": <function-name>, "arguments": <args-json-object> }
1011    </tool_call>
1012    You may call one or more functions to assist with the user query.
1013
1014    # How to call a turn summary:
1015    Return a JSON object with name and arguments within <turn_sum></
1016        turn_sum> XML tags:
        <turn_sum>
1017    { "name": "TurnSum", "arguments": { "attempt": "...", "observation":
1018        "...", "status": "need_more_info|partial_progress|blocked", "
1019        next_step": "..." } }
        </turn_sum>
1020
1021    # Output Protocol (STRICT)
1022    - Per-turn pattern: Tool turn -> <think>...</think> ( <tool_call
1023        >{...}</tool_call> ){1,3}<turn_sum>...</turn_sum>;
        - Every turn MUST summary the content of the turn and end with a
1024        closed <turn_sum>... </turn_sum>.
1025
```

```
USER_PROMPT(Turn 1)

Start with <think>.
Format strictly as: <think>...</think> <tool_call>...</tool_call> <
turn_sum>...</turn_sum>.
Please think about this question as if you were a human pondering
deeply. Its encouraged to include self-reflection or verification in
 the reasoning process. Provide your detailed reasoning between the
<think> and </think> tags. All your formal output should be a brief
sentence, less than 100 tokens.
You MUST end in <turn_sum>...</turn_sum>. Use <tool_call> to get the
 specific segments of videos or frames.
```

```
TURN_PROMPT(Turn 2,3...)

<tool_response>{visual_content}</tool_response>
Based on the tool response above, analyze the video content and
answer the original question.
Start with <think> and analyze the time information and the content
shown in these frames.
You can use <tool_call> if you need to get the specific segments of
videos or frames.
If the information is enough, output your answer after <answer> and
end in </answer>. If not, summary the content of the turn after <
turn_sum> and end in </turn_sum>.
```

### E.2 TRAINING DETAILS

We train a tool-augmented video understanding model based on Qwen/Qwen2.5-VL-7B-Instruct. Optimization uses AdamW (lr $1 \times 10^{-6}$, weight decay $1 \times 10^{-2}$), no warmup, and gradient clipping at 1.0, with gradient checkpointing enabled. We employ FSDP full sharding (no CPU offload) and tensor parallel size 4. Batching uses a global batch of 64 with per-device micro-batches of 2 for updates and 1 for experience; padding-free and dynamic batching are enabled. RL follows GRPO with a KL regularizer (low_var_kl, coefficient $10^{-3}$). The visual budget is constrained by min_pixels = 50176 and max_pixels = 200704, and overlong prompts are kept. For rollouts we use DRFS multi-view with $n = 8$ (from lower-res/more-frames to higher-res/fewer-frames), temperature 0.7, top-$p = 0.9$, up to 50 images per prompt, and target GPU memory utilization 0.7. Context limits are max_model_len = 32768 and max_num_batched_tokens = 65536. For validation we adopt conservative settings: temperature 0.01, top-$p = 0.95$, single view $n = 1$, and max_frames = 64. Key settings are summarized in Table E1.

### E.3 REWARD MANAGER

The Reward Manager is a crucial component responsible for calculating the accuracy component of the final reward signal, $R_{acc}(\tau)$. To handle the diverse nature of video question-answering tasks, our training framework employs two distinct scoring methodologies based on the type of question being evaluated.

- **Exact Match (EM) Scoring**: For questions that have a definitive, single correct answer, such as numerical questions or multiple-choice questions, we use an Exact Match (EM) scoring method. The model's generated answer is compared directly against the ground truth. A reward is granted only if the answer is a perfect match. This provides a clear and objective signal for convergent tasks.

- **LLM as Judge Scoring**: For open-ended and descriptive questions, where answers can be semantically correct but vary in phrasing, a simple string comparison is inadequate. For these cases, we utilize an **LLM as Judge** to provide a more nuanced reward.

We specifically use **GPT-4o mini** as the judge for its strong reasoning capabilities and efficiency. The judge model is provided with the question, the ground truth answer, and the model-generated answer, and it assesses the quality based on the following structured prompt:

```
LLMasJudge_PROMPT

You are an expert evaluating video understanding accuracy for free-
form questions. Below are two answers to a video question: [Question
] is the task, [Standard Answer] is correct, and [Model_answer] is
the model's response.

**General Evaluation Principles:**
- Both answers should demonstrate understanding of the same video
content
- Accept different wording, style, and organization if meaning is
equivalent
- Be lenient with minor details but strict with major factual errors
- Consider the overall coherence and completeness of understanding
- Focus on whether both answers would be considered correct by a
human evaluator

**Scoring Guidelines:**
- Score 1 if answers show equivalent video understanding despite
different expression
- Score 0 if answers show fundamentally different understanding of
the video content
- Be generous with semantic equivalence but strict with factual
accuracy

If the video understanding is consistent, output Judgement: 1; if
different, output Judgement: 0.
```

### E.4 ANALYSIS OF TRAINING PARADIGMS AND DATA EFFICIENCY

In this section, we provide a detailed comparison of the training methodologies, data modalities, and data volumes used by FrameMind versus other state-of-the-art models, as summarized in Table E2.

A key distinction of our work lies in the **training paradigm**. The majority of contemporary video MLLMs, such as Video-CCAM, VideoChat2, and LongVA, rely on a standard Supervised Fine-Tuning (SFT) paradigm. Video-R1 employs a hybrid approach, using a large SFT stage followed by an RL phase. In contrast, FrameMind is fine-tuned using a pure **Reinforcement Learning (RL)** approach, where the model learns a reasoning policy from outcome-based rewards rather than explicit instruction-response pairs.

This difference in paradigm is also reflected in the **data modalities**. Most baselines are trained on a combination of image and video data ('img/vid-text'), whereas FrameMind's training is focused exclusively on 'vid-text' data.

The most significant finding from this comparison is the remarkable **data efficiency** of our framework. FrameMind is trained on a curated dataset of only **7.6K** instances. This is one to three orders of magnitude smaller than the fine-tuning datasets used by other models, which range from **257K** for Video-XL to **4.4M** for Video-CCAM. For a direct comparison, FrameMind uses approximately 34 times less data than the SFT+RL model, Video-R1. This result strongly suggests that our agentic FiCOT process, trained with DRFS-GRPO, learns a more generalizable and sample-efficient reasoning policy from sparse rewards, avoiding the need for massive, and often costly, supervised datasets.

## F CASE STUDY

Table E1: Training Config.

| Configuration | Value |
|---|---|
| **Data constraints** | |
| min_pixels / max_pixels | `50176 / 200704` |
| **Algorithm** | |
| adv_estimator | `grpo` |
| use_kl_loss / kl_penalty / kl_coef | `true / low_var_kl / 0.001` |
| disable_kl / online_filtering | `false / false` |
| **Model** | |
| model_path | `Qwen/Qwen2.5-VL-7B-Instruct` |
| freeze_vision_tower | `false` |
| enable_gradient_checkpointing | `true` |
| tensor_parallel_size | `4` |
| **Batching** | |
| global_batch_size | `64` |
| micro_batch_size_per_device_for_update | `2` |
| micro_batch_size_per_device_for_experience | `1` |
| padding_free / dynamic_batching | `true / true` |
| max_grad_norm | `1.0` |
| ulysses_size | `1` |
| **Optimization** | |
| strategy | `adamw` |
| lr / weight_decay / lr_warmup_ratio | `1.0e-6 / 1.0e-2 / 0` |
| **FSDP & Offload** | |
| fsdp: full_shard / cpu_offload / rank0_init | `true / false / true` |
| offload_params / offload_optimizer | `false / false` |
| **Rollout / Inference** | |
| n (DRFS views) | `8` |
| $\max_t urns$ | `3` |
| temperature / top_p | `0.7 / 0.9` |
| limit_images / gpu_memory_utilization | `100 / 0.7` |
| enforce_eager / chunked_prefill / disable_tqdm | `false / false / false` |
| max_model_len / max_num_batched_tokens | `32768 / 65536` |
| val_override: temperature / top_p / n / max_frames | `0.01 / 0.95 / 1 / 64` |

| Method | Size | Paradigm | Modalities | Volume |
|---|---|---|---|---|
| Video-CCAM | 9B | SFT | img/vid-text | 4.4M |
| VideoChat2 | 7B | SFT | img/vid-text | 2M |
| LongVA | 7B | SFT | img-text | 1.3M |
| Video-XL | 7B | SFT | img/vid-text | 257K |
| Video-R1 | 7B | SFT+RL (S) | img/vid-text | 260K |
| FrameMind(Ours) | 7B | RL(M) | vid-text | 7.6K |

Table E2: Comparison of training paradigms, data modalities and volumes. (M)/(S) denote multi-turn and single-turn respectively. All SFT implementations are instruction tuning. For a strictly fair comparison, we only compare the data used during the fine-tuning stage.

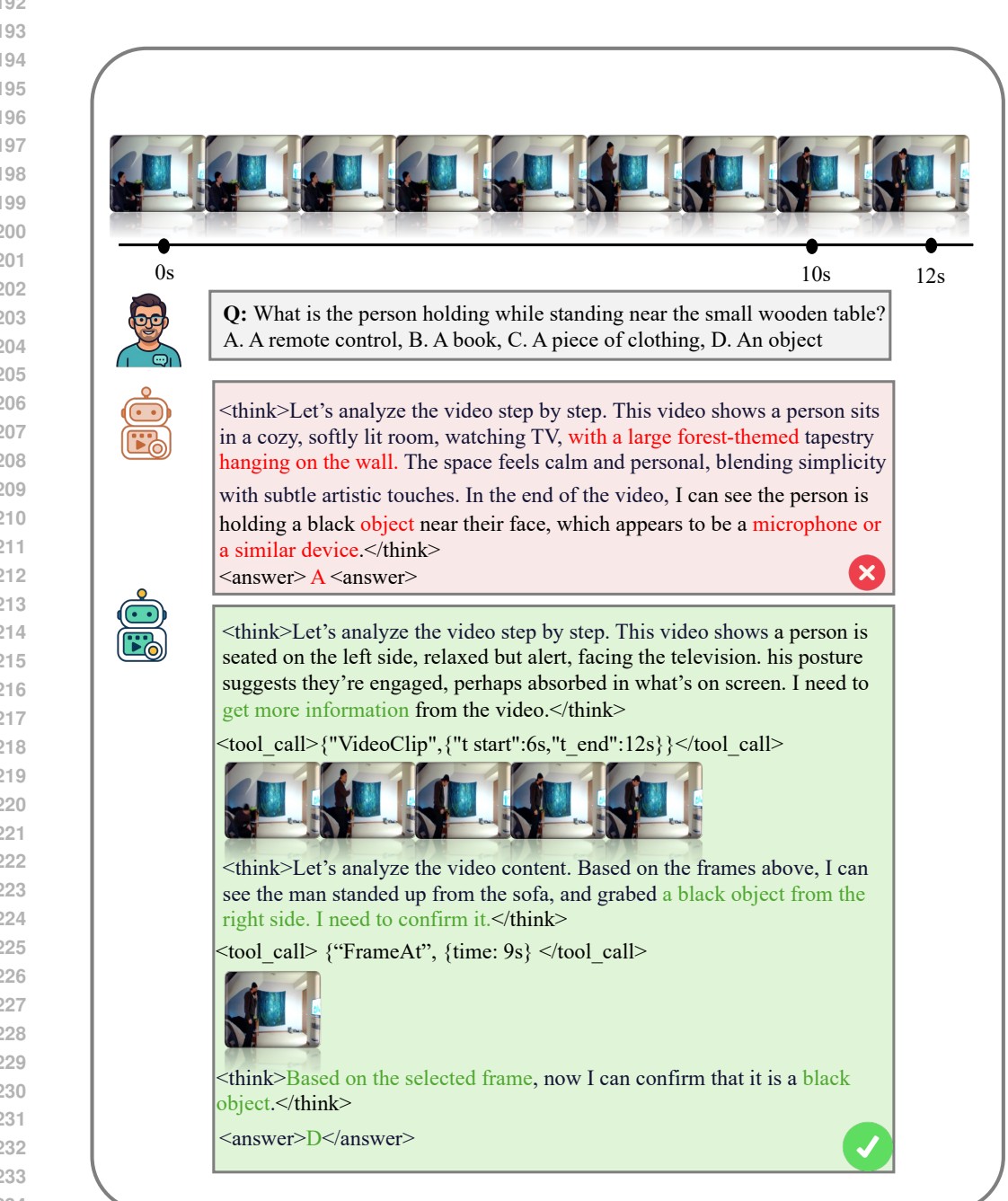

Figure F1: FrameMind Sample 1.

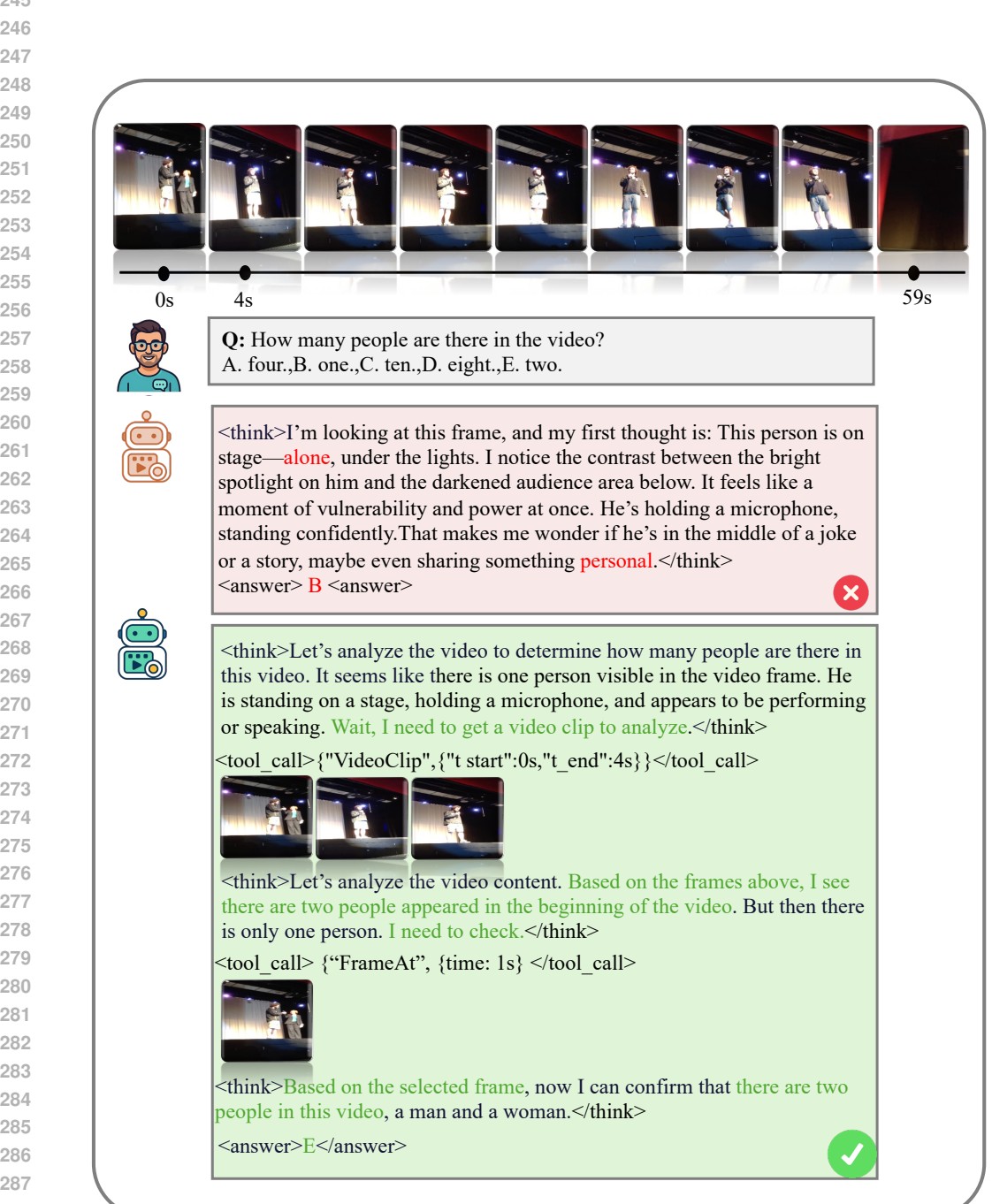

Figure F2: FrameMind Sample 2.

