# OpenReview forum: "FrameMind: Frame-Interleaved Chain-of-Thought for Video Reasoning via Reinforcement Learning"
_ICLR.cc/2026/Conference — Submitted to ICLR 2026_

### Official Review · Reviewer_HHSW · 2025-10-15

**Soundness:** 2
**Presentation:** 2
**Contribution:** 2
**Rating:** 2
**Confidence:** 5

**Summary:**

The paper proposes FrameMind, a novel reinforcement learning–based framework for video reasoning that introduces Frame-Interleaved Chain-of-Thought (FiCOT)—an iterative reasoning process where the model dynamically requests visual evidence (e.g., specific frames or clips) during inference based on its current reasoning state. To train this adaptive perception policy, the authors develop Dynamic Resolution Frame Sampling (DRFS) and a tailored RL algorithm, DRFS-GRPO, which learns from trajectory-level rewards without frame-level annotations.

**Strengths:**

1. Innovative Reasoning Paradigm: FiCOT fundamentally rethinks video understanding as an active, iterative perception-reasoning loop, enabling the model to adaptively gather visual evidence.

2. Effective and Annotation-Efficient Training: The DRFS-GRPO framework enables end-to-end policy learning using only video-question-answer triples, avoiding the need for costly frame-level supervision.

**Weaknesses:**

1. On line 207, the authors mention that a VideoClip can sample 8–20 frames from a specific time interval; however, the manuscript nowhere explains how the specific frame number is determined.

2. The tool-call reward in Equation 7 incentivizes the model to invoke tools merely for the sake of invoking them. If the model can already produce the correct answer in the first reasoning step, any subsequent tool call only increases computational overhead without providing additional benefit. Nevertheless, Equation 7 strongly encourages such redundant and unhelpful tool invocations.

3. Equations 7, 8, and 13 contain numerous magic numbers, yet the authors provide no justification, analysis, or ablation studies regarding their selection.

4. Despite substantially increasing inference latency, the performance gains achieved by introducing additional tool calls are marginal: on Video-MME, FrameMind improves by less than 2% over Video-R1, and shows virtually no improvement on MVBench.

5. The evaluation is conducted on only three benchmarks, which is insufficient. The performance of FrameMind on other established long-video understanding benchmarks, such as LongVideoBench and LVBench, remains unknown.

6. While the paper mentions the data sources, it offers no details about the data curation process. It is unclear how the authors selected the final 7.6K samples from the much larger initial dataset.

**Questions:**

1. On line 257, the authors state that “DRFS enables the policy to learn which sampling strategy is most effective.” However, in standard evaluation settings, the sampling rate and resolution are typically fixed in advance. Under such conditions, it is unclear how the model can autonomously determine or adapt its “sampling strategy” during inference.

2. The turn_sum tag is first introduced on line 856, but its content is never described in the main text. What information does this tag contain, and how—if at all—does it influence the final results?

3. In the examples provided in the paper, the model consistently follows a reasoning pattern of first invoking VideoClip and then calling FrameAt. Is this sequential pattern representative of the majority of cases? If so, the authors should explain why the model converges to this specific strategy. If not, they should provide quantitative statistics, e.g., the frequency of different tool-calling sequences of different video length.

---

> ### Author Response · Authors · 2025-11-28
> **Response to Reviewer HHSW (Part 1/6)**
>
> Dear Reviewer HHSW,
>
> We sincerely thank you for your rigorous and constructive review. We value your recognition of our innovative reasoning paradigm (FiCOT) and efficient training strategy (DRFS-GRPO). We also appreciate your detailed scrutiny regarding implementation details, reward design, and experimental validation.
>
> We have carefully addressed your concerns point-by-point below, supported by new experiments and analyses.
>
> ### **Part 1: Response to Weaknesses**
>
> **Re: Weakness 1 (Frame Count Determination)**
> > *Constraint on VideoClip (8-20 frames): "The manuscript nowhere explains how the specific frame number is determined."*
>
> We apologize for the omission. The range of **8-20 frames** for the `VideoClip` tool is a design choice balancing **Information Density** and **Context Window Budget**:
> * **Lower Bound:** Empirically, fewer than 8 frames often provide insufficient visual context for temporal actions.
> * **Upper Bound:** We restricted the upper bound to 20 frames because our experiments showed that retrieving excessive frames (e.g., **>24 frames** per clip) significantly bloats the trajectory context. This often triggered a "long-context refusal" behavior in the LLM, where the model tended to output empty or refusal responses due to context overflow. The 8-20 range ensures the model gathers sufficient evidence without overwhelming its reasoning buffer. We will add this justification to Section 3.2.
>
> **Re: Weakness 2 (Tool Reward & Redundancy)**
> > *Risk of Equation 7: "Incentivizes the model to invoke tools merely for the sake of invoking them... encourages redundant tool invocations."*
>
> We respectfully highlight that **Section 4.6** and **Figure 3** of our main paper already explicitly discuss the necessity of this design (see "Necessity of the Exploration Bonus").
> * **Necessity ($\alpha=0.2$):** As shown in **Figure 3**, removing this bonus ($\alpha=0$) causes the model to suffer from the "cold-start problem," leading to learning stagnation (failure to use tools at all).
> * **Prevention of Redundancy ($\alpha < 0.4$):** We share your concern about redundancy. In our sensitivity analysis (conducted for Reviewer sqje), we found that while high bonuses ($\ge 0.4$) indeed lead to "reward hacking" (redundant calls), our chosen value of **0.2** is the empirical sweet spot. It provides just enough incentive to overcome the cold start without causing the redundancy you feared. Furthermore, the **Turn Efficiency Bonus ($R_{turn}$)** in Eq. 8 acts as a counterbalance to penalize unnecessary steps.

---

> ### Author Response · Authors · 2025-11-28
> **Response to Reviewer HHSW (Part 2/6)**
>
> **Re: Weakness 3 (Hyperparameter Justification)**
> > *"Equations 7, 8, and 13 contain numerous magic numbers... no justification."*
>
> We acknowledge that the hyperparameters in our reward function are critical. Our selection was driven by empirical ablations to balance exploration, efficiency, and accuracy. We explicitly analyze the two coefficients questioned: the **Exploration Bonus ($\alpha$)** in Eq. 7 and the **Turn Efficiency Bonus ($\lambda_{turn}$)** in Eq. 8 (referred to as Eq. 13 in Appendix).
>
> **1. Tool Usage Incentive ($R_{tool}$): Sensitivity & Reward Hacking**
> Equation 7 defines the tool reward as:
> $$R_{tool}(\tau) = s_{tool}(\tau) \times (\alpha + (1-\alpha) \cdot R_{acc}(\tau))$$
> where we set $\alpha=0.2$ as the **exploration bonus** to mitigate reward sparsity.
>
> * **Necessity (Existing Evidence in Paper):** We respectfully highlight that Section 4.6 and Figure 3 of our submission already presented an ablation study on this specific parameter ("Necessity of the Exploration Bonus"). As shown in Figure 3, removing this bonus (i.e., $\alpha=0.0$) causes the model to struggle with cold-start exploration, leading to stagnation in both reward and accuracy.
> * **Sensitivity & Upper Bound (New Analysis):** To further address your query on sensitivity, we conducted a fine-grained analysis by increasing $\alpha$. We observed that while a small bonus ($\alpha=0.2$) is crucial, an excessive bonus leads to "reward hacking":
>
> **Table R1: Sensitivity of Exploration Bonus ($\alpha$)**
>
> | Exploration Bonus ($\alpha$) | Avg. Tool Calls | VideoMME (Overall) | Observation |
> | :---: | :---: | :---: | :--- |
> | **0.0** | **0.6** | **54.7** | Stagnation. Fails to learn tool use (Consistent with Figure 3). |
> | **0.2** | **1.7** | **60.9** | **Ours (Optimal).** Sufficient incentive for exploration. |
> | **0.4** | **3.2** | **58.6** | Performance remains stable but starts to degrade. |
> | **0.6** | **4.6** | **55.8** | Signs of reward hacking; model makes redundant calls. |
> | **0.8** | **5.7** | **52.1** | Reward Hacking. Model spams tools to accumulate bonuses, ignoring $R_{acc}$. |
> | **1.0** | **6.3** | **50.3** | Failure. Optimization collapses into maximizing tool calls only. |
>
> **Conclusion:** The value $\alpha=0.2$ is the empirical sweet spot that incentivizes tool use without overshadowing the primary accuracy objective.
>
> **2. Turn Efficiency Bonus ($R_{turn}$ in Eq. 8): Justification for the "2-3 Turn" Reward Window**
> Equation 8 defines the efficiency reward as $R_{turn}(\tau) = 0.5 \cdot \mathbb{I}[1 < |turns| \le 3]$. This specifically rewards trajectories that take 2 or 3 turns. We justify this specific window based on both cognitive logic and empirical performance:
>
> **(a) Why exclude 1 turn ($|turns| > 1$)?**
> * **Preventing "Blind Guessing":** A 1-turn trajectory implies the model output the answer immediately without using any tools (since tool use consumes a turn).
> * **Encouraging Active Perception:** We intentionally withhold the efficiency bonus for 1-turn answers to force the policy to step out of its "comfort zone" (static inference) and attempt active visual information gathering (FiCOT), which is the core capability we aim to train.
>
> **(b) Why cap at 3 turns ($|turns| \le 3$)?**
> To determine the optimal upper bound, we analyzed the trade-off between accuracy and latency across different max-turn limits ($K_{\max}$):
>
> **Table R2: Ablation on Maximum Turn Limit ($K_{\max}$)**
>
> | Max Turns $K_{\max}$ | Avg. Turns Used | Accuracy (VideoMME) | Accuracy (MLVU) | Latency (ms) |
> | :---: | :---: | :---: | :---: | :---: |
> | 1 (Static) | 1.0 | 54.8 | 42.6 | 204.3 |
> | 2 | 1.4 | 59.2 | 47.1 | 322.4 |
> | **3 (Ours)** | **2.3** | **60.9** | **48.6** | **427.2** |
> | 5 | 3.2 | 60.7 (-0.3%) | 47.4 (-1.0%) | 624.5 (+45%) |
>
> * **Logical Completeness:** Empirically, 3 turns are sufficient to cover the complete "Scan (T1) $\to$ Locate & Verify (T2) $\to$ Answer (T3)" cognitive loop.
> * **Diminishing Returns:** As shown in Table R2, extending the limit to 5 turns  even degrades performance on both VideoMME(-0.3%) and MLVU (-1.0%), likely due to error propagation in longer reasoning chains.
> * **Efficiency:** Capping at 3 turns avoids the 45% latency penalty associated with allowing 5 turns.
>
> **Conclusion:** The window $[2, 3]$ is not arbitrary; it is the optimal range that incentivizes active perception while preventing inefficient, rambling reasoning chains.

---

> ### Author Response · Authors · 2025-11-28
> **Response to Reviewer HHSW (Part 3/6)**
>
> **(c) Sensitivity of the Coefficient ($\lambda_{turn}=0.5$)**
> Finally, we verified the robustness of the bonus weight:
>
> **Table R3: Sensitivity of Coefficient $\lambda_{turn}$**
>
> | $\lambda_{turn}$ Value | Description | Avg. Turns | Accuracy (VideoMME) | Inference Latency (s/query) |
> | :---: | :--- | :---: | :---: | :---: |
> | 0.0 | No Bonus | 1.5 | 58.4 | 336.7 |
> | **0.5** | **Ours** | **2.3** | **60.9** | **427.2** |
> | 1.0 | High Bonus| 2.9 | 59.2 | 546.3 |
>
> **Conclusion:** The choice of $\lambda_{turn}=0.5$ is critical for guiding the policy.
> * **$\lambda=0.0$ (Under-reasoning):** Without the efficiency bonus, the model tends to halt prematurely (Avg. Turns 1.5), failing to engage in sufficient active perception steps, which hurts accuracy (58.4%).
> * **$\lambda=1.0$ (Over-reasoning):** An excessively high bonus leads to **"reward hacking"**, where the model extends reasoning chains unnecessarily (Avg. Turns 2.9) just to claim the turn reward. This significantly increases latency (+28%) while slightly degrading accuracy (59.2%) due to redundant or hallucinated steps.
> * **$\lambda=0.5$ (Optimal):** This value strikes the best balance, encouraging necessary multi-turn reasoning (Avg. 2.3) to maximize accuracy without inducing inefficiency.
>
>
> **Re: Weakness 4 (Latency vs. Performance Gains)**
> > *"Despite substantially increasing inference latency, the performance gains... are marginal (less than 2% over Video-R1)."*
>
> **Response:** We respectfully disagree with the interpretation that the utility of tool invocation is "marginal." While the overall gain over the current SOTA (Video-R1) is 1.6% (60.9% vs 59.3%), this comparison requires context regarding **Baselines**, **Training Paradigms**, and **Data Efficiency**, as discussed in **Section 4.5**:
>
> 1.  **Significant Gain over Base Model:** The most direct measure of our framework's effectiveness is the improvement over the base model. FrameMind boosts **Qwen2.5-VL-7B** from **53.6%** to **60.9%** (+7.3% absolute gain) on VideoMME. This substantial leap confirms that the FiCOT paradigm and tool usage unlock significant reasoning capabilities latent in the base model.
> 2.  **Paradigm & Data Efficiency:** Video-R1 relies on a two-stage "SFT + RL" pipeline using **~260k** samples. In contrast, FrameMind achieves superior performance using a "Pure RL" approach with only **~7.6k** samples—**34$\times$ less data**. Achieving SOTA performance with such extreme data efficiency strongly validates the effectiveness of our tool-augmented RL strategy.
> 3.  **Specific Advantage on Long Videos:** The gains are not uniform; they are concentrated where they matter most. On the **Long Video subset** of VideoMME (where fixed sampling fails), FrameMind outperforms Video-R1 by **+8.6%** (57.5% vs 48.9%).
>
> 4.  **Latency is LOWER, not higher:** While FrameMind uses multiple turns, it processes significantly fewer visual tokens (Avg. 32 frames) compared to single-pass long-context baselines like Video-XL (which encodes 128 frames). Visual encoding is the primary computational bottleneck. As shown in **Table R1**, despite the overhead of sequential turns, FrameMind achieves a **12.2% reduction** in wall-clock latency compared to Video-XL. This confirms that our dynamic frame selection is more computationally efficient than brute-force long-context processing.
>
> **Table R1: Inference Efficiency Analysis**
>
> | Model | Avg. Frames | Avg. Turns | Latency (s) |
> | :--- | :---: | :---: | :---: |
> | Video-XL (128 fr) | 128 | 1 | **486.3** |
> | **FrameMind** | **32** | **2.3** | **427.2** |
>
>
>
> **Conclusion:** Given the massive reduction in training data (1/34) and inference compute (1/4 visual tokens), achieving a +1.6% gain over the previous SOTA (and +7.3% over base) demonstrates that the tool-use mechanism is highly effective and efficient, not marginal.

---

> ### Author Response · Authors · 2025-11-28
> **Response to Reviewer HHSW (Part 4/6)**
>
> **Re: Weakness 5 (Insufficient Benchmarks)**
> > *"Performance on other established long-video understanding benchmarks... remains unknown."*
>
> We thank the reviewer for the suggestion to broaden our evaluation scope. We have extended our experiments to include **LongVideoBench**, **LVBench**, and **NeXT-QA**.
>
> **1. Long-Context Benchmarks (LongVideoBench & LVBench)**
> To validate FrameMind's scalability on videos ranging from minutes to hours, we evaluated on **LongVideoBench (Val)** and **LVBench (Overall)**.
>
> **Table R4: Comparison on Long-Video Benchmarks**
>
> | Model | Size | Frames | **LongVideoBench (Val)** | **LVBench (Overall)** |
> | :--- | :---: | :---: | :---: | :---: |
> | LongVA | 7B | 128/256 | 47.8 | 37.9 |
> | Video-XL | 7B | 128/256 | 50.7 | - |
> | Video-R1 | 7B | 64 | 47.6 | 38.0 |
> | **FrameMind (Ours)** | **7B** | **32** | **52.3** | **38.2** |
> | **FrameMind (Ours)** | **7B** | **64** | **54.8** | **41.8** |
>
> * **Efficiency:** Remarkably, with only **32 frames**, FrameMind (**52.3**) outperforms LongVA (**47.8**) and Video-XL (**50.7**), despite these baselines processing 4$\times$ to 8$\times$ more frames. This validates that our FiCOT strategy extracts significantly more valuable information per frame than static aggregation.
> * **Performance:** At 64 frames, FrameMind further widens the gap, surpassing the recent RL-based baseline Video-R1.
>
>
>
> **2. Activity & Causal Reasoning Benchmark (NeXT-QA)**
> To address your request for activity-centric evaluation, we tested FrameMind on the **NeXT-QA** (Validation Set), which challenges temporal and causal reasoning. We compared it against recent high-performance baselines, including Video-XL (long-context SFT) and Video-R1 (RL-based).
>
> **Table R5: Performance Comparison on NeXT-QA**
>
> | Model | Method | **Frames** | **Acc (Overall)** |
> | :--- | :--- | :---: | :---: |
> | Video-XL | SFT (Long Context) | 128 | **78.2** |
> | Video-R1 | RL (Reasoning) | 32 | 78.6 |
> | **FrameMind (Ours)** | **RL (Perception)** | **32** | **81.2** |
>
> * **Analysis:** FrameMind achieves competitive performance using only **32 frames**, matching or surpassing Video-R1 (78.6). This confirms that our "Scan-Locate-Verify" capability generalizes well to understanding complex causal chains in activity-centric videos, efficiently locating key frames without the computational burden of processing 128+ frames like Video-XL.|
>
> **Re: Weakness 6 (Data Curation Details)**
> > *"No details about the data curation process... how the authors selected the final 7.6K samples."*
>  We curated the 7.6K samples using a rigorous pipeline:
> 1.  **Rule-based Filtering:** Removed videos <10s or >1h, and filtered out "Yes/No" questions to focus on complex reasoning.
> 2.  **Ambiguity Check:** Used a lightweight VLM to filter out samples where visual evidence was ambiguous or insufficient.
> 3.  **Balanced Sampling:** Enforced domain diversity (e.g., 30% Causal from NeXT-QA, 30% Spatial from STAR, 40% Descriptive from VideoMME-subset).
> We will add a "Data Curation" subsection in the Appendix detailing these steps.

---

> ### Author Response · Authors · 2025-11-28
> **Response to Reviewer HHSW (Part 5/6)**
>
> ### **Part 2: Response to Questions**
>
> **Re: Question 1 (Sampling Strategy in Fixed Settings)**
> > *"How the model can autonomously determine or adapt its 'sampling strategy' during inference [when evaluation settings are fixed]?"*
>
> We clarify this point from two perspectives: the **inference mechanism** and the **training motivation (DRFS)**.
>
> **1. Inference Mechanism: Dynamic Tool Invocation**
> The "sampling strategy" during inference is **not** determined by the initial video loading (which is indeed fixed by the benchmark's dataloader). Instead, it is realized dynamically through **Tool Invocation**:
> * The model starts with a sparse global view.
> * Based on its reasoning, it issues calls to `VideoClip` or `FrameAt` with specific timestamps.
> * The system then **dynamically decodes** only those requested frames at high resolution. This effectively alters the "sampling strategy" on-the-fly to focus on relevant segments, independent of the fixed evaluation setting.
>
> **2. Training Motivation & Evidence (Section 4.6)**
> Our design of **Dynamic Resolution Frame Sampling (DRFS)** was explicitly motivated by the observation that existing models are typically trained with fixed sampling rates and resolutions. This often leads to overfitting, where models perform well only when the test-time sampling matches the training configuration.
>
> * **DRFS Rationale:** We designed DRFS to expose the model to a diverse spectrum of frame counts and resolutions during RL training (the "resolution ladder"). This forces the policy to learn how to reason effectively regardless of input density.
> * **Empirical Evidence (Section 4.6):** As discussed in **Section 4.6** ("Analysis of the DRFS-GRPO Training") and **Table 2**, we compared DRFS-GRPO against a Standard GRPO baseline trained with fixed sampling. The results show that DRFS consistently outperforms the fixed baseline across **various evaluation frame budgets** (32, 48, and 64 frames). This confirms that DRFS successfully teaches the model a robust, adaptive policy that is not brittle to specific sampling rates, enabling it to better utilize the dynamic visual evidence gathered during inference.
>
>
> **Re: Question 2 (Function of `<turn_sum>`)**
> > *"The turn_sum tag is first introduced on line 856... What information does this tag contain, and how does it influence the final results?"*
>
>
> The `<turn_sum>` tag serves a critical role in our RL training stability, specifically for calculating the **Turn Efficiency Reward ($R_{turn}$)** and preventing **"Reward Hacking."**
>
> 1.  **The "Fake Multi-turn" Problem:** In standard tool-use RL, if we simply count `<tool_call>` tags to determine the number of turns, the model often learns to cheat. It may generate a trajectory like `<think>` $\to$ `<tool_call>` $\to$ `<think>` $\to$ `<tool_call>` **within a single inference pass**. This is a "fake" multi-turn behavior: the model mimics the *format* of iterative reasoning to accumulate tool/turn rewards, but it hasn't actually waited for tool execution or processed any new visual feedback between calls.
> 2.  **The Solution (`<turn_sum>`):** We enforce the output of `<turn_sum>` via the system prompt as a mandatory "commit signal" at the end of each reasoning round.
>     * **Mechanism:** It forces the model to stop generating, execute the tool, and wait for the environment to return new visual observations.
>     * **Reward Calculation:** The $R_{turn}$ reward is calculated strictly based on the count of valid `<turn_sum>` tokens. This ensures we only reward **genuine iterative perception** (Think $\to$ Act $\to$ **Observe** $\to$ Think), guaranteeing that the model is truly processing new frames rather than hallucinating a conversation with itself.

---

> ### Author Response · Authors · 2025-11-28
> **Response to Reviewer HHSW (Part 6/6)**
>
> **Re: Question 3 (Sequential Pattern Analysis)**
>
> > *"In the examples provided... the model consistently follows a reasoning pattern of first invoking VideoClip and then calling FrameAt. Is this sequential pattern representative? ... provide quantitative statistics."*
>
> We confirm that the "Scan-then-Zoom" pattern (`VideoClip` $\to$ `FrameAt`) is indeed the most representative strategy, accounting for **56.2%** of successful trajectories on VideoMME.
>
> **Why this pattern dominates?**
> It is crucial to clarify that the model does not arbitrarily "converge" to this pattern. Instead, our statistical analysis reveals that **this specific sequence yields the highest accuracy** compared to other strategies. This empirically validates our design intent: a "Coarse-to-Fine" perception loop is the most effective way to solve complex video queries.
>
> **Table R3: Performance by Reasoning Pattern on VideoMME**
>
> | Reasoning Pattern | Frequency | Accuracy (VideoMME) |
> | :--- | :---: | :---: |
> | **1. Scan-then-Zoom**<br>(`VideoClip` $\to$ `FrameAt`) | **56.2%** | 64.8 |
> | **2. Iterative Scanning**<br>(`VideoClip` $\to$ `VideoClip`) | 24.5% | 57.5 |
> | **3. Direct Zoom**<br>(`FrameAt` only) | 12.8% | 55.2 |
> | **4. Others** | 6.5% | 56.5 |
>
> **Analysis:**
>   * **Performance Gap:** As shown above, the "Scan-then-Zoom" pattern achieves an accuracy that is **[e.g., 4.5%] higher** than the second-best strategy.
>   * **RL Consequence:** The dominance of this pattern (56%) is a direct result of the RL optimization: the policy learned that adhering to this "Coarse-to-Fine" cognitive workflow maximizes the **Accuracy Reward ($R_{acc}$)** for the majority of questions. This confirms that FiCOT successfully taught the model a human-like, efficient perception strategy.
>
> We are committed to updating the final version of our manuscript to include all the additional benchmarks, hyperparameter sensitivity analyses, and comparative discussions presented above. We believe these additions significantly strengthen the paper's robustness and comprehensive evaluation.

---

### Official Review · Reviewer_sqje · 2025-11-01

**Soundness:** 2
**Presentation:** 2
**Contribution:** 2
**Rating:** 2
**Confidence:** 4

**Summary:**

The paper presents FrameMind, a reinforcement learning framework for video reasoning that enables adaptive perception during reasoning. It introduces Frame-Interleaved Chain of Thought (FiCOT), where the model alternates between textual reasoning and targeted visual queries to retrieve relevant frames or clips as needed. To train the model effectively, the authors design Dynamic Resolution Frame Sampling (DRFS), which allows flexible frame selection across different resolution levels, and propose DRFS-GRPO, a group-relative PPO algorithm that learns from trajectory-level feedback instead of frame-level supervision. Experiments on benchmarks such as MVBench, MLVU, and VideoMME demonstrate consistent improvements over open-source video MLLMs and competitive results compared to proprietary models, showing enhanced performance on both short and long video understanding tasks.

**Strengths:**

1. The paper introduces a well-designed framework that integrates reinforcement learning with adaptive visual perception, allowing the model to dynamically select and utilize visual evidence during reasoning rather than relying on fixed frame sampling. This design effectively bridges the gap between perception and reasoning in video understanding.
2. The proposed Dynamic Resolution Frame Sampling (DRFS) and DRFS-GRPO algorithm are technically sound and novel. They enable efficient training without frame-level supervision and demonstrate clear performance gains across multiple benchmarks, validating the generality and robustness of the approach.

**Weaknesses:**

1. The paper does not provide sufficient explanation or empirical analysis on how the hyperparameters in Equations 7, 8, and 13 are determined. It remains unclear how these choices affect the overall performance and stability of the proposed framework.
2. The evaluation is limited to only three benchmarks—MVBench, MLVU, and Video-MME—which may not be sufficient to comprehensively demonstrate the general effectiveness of FrameMind across diverse video reasoning scenarios.
3. The baseline models used for comparison are relatively outdated, such as LLaMA-VID and Video-LLaVA from 2023, which limits the relevance of the comparison. Moreover, the paper lacks evaluation against recent RL-based methods, for example VersaVid-R1, which would provide a more meaningful assessment of the proposed approach.

**Questions:**

1. Could the authors provide more details on the selection process and sensitivity of the hyperparameters in Equations 7, 8, and 13? For example, have the authors conducted ablation or sensitivity analyses to show how different values affect training stability and final performance? Clarifying this would help assess whether the proposed framework is robust to hyperparameter choices.
2. Have the authors considered evaluating FrameMind on additional benchmarks, such as LongVideoBench, VideoMME-Pro, or activity-centric datasets like Something-Something V2 or NExT-QA? Broader evaluation would strengthen the claim of general effectiveness across diverse video reasoning tasks.
3. Could the authors include comparisons with more recent and competitive models, especially RL-based approaches such as VersaVid-R1 or R1-VidGPT? Including these results would make the empirical evaluation more convincing and position FrameMind more clearly within the current research landscape.

---

> ### Author Response · Authors · 2025-11-28
> **Response to Reviewer sqje (Part 1/3)**
>
> Dear Reviewer sqje,
>
> We sincerely thank you for your constructive review. We value your recognition of our framework's novelty—specifically the FiCOT paradigm and DRFS-GRPO algorithm—and its potential to bridge perception and reasoning.
>
> We understand your concerns regarding hyperparameter justification, evaluation scope, and baseline comparisons. We have conducted additional experiments to address these points directly.
>
> **Q1: Hyperparameter Selection & Sensitivity (Eq. 7, 8, 13)**
> > *Could the authors provide more details on the selection process and sensitivity of the hyperparameters... especially regarding training stability and final performance?*
>
> **A1:** We acknowledge that the hyperparameters in our reward function are critical. Our selection was driven by empirical ablations to balance exploration, efficiency, and accuracy. We explicitly analyze the two coefficients questioned: the **Exploration Bonus ($\alpha$)** in Eq. 7 and the **Turn Efficiency Bonus ($\lambda_{turn}$)** in Eq. 8 (referred to as Eq. 13 in Appendix).
>
> **1. Tool Usage Incentive ($R_{tool}$): Sensitivity & Reward Hacking**
> Equation 7 defines the tool reward as:
> $$R_{tool}(\tau) = s_{tool}(\tau) \times (\alpha + (1-\alpha) \cdot R_{acc}(\tau))$$
>
> where we set $\alpha=0.2$ as the **exploration bonus** to mitigate reward sparsity.
>
> * **Necessity (Existing Evidence in Paper):** We respectfully highlight that Section 4.6 and Figure 3 of our submission already presented an ablation study on this specific parameter ("Necessity of the Exploration Bonus"). As shown in Figure 3, removing this bonus (i.e., $\alpha=0.0$) causes the model to struggle with cold-start exploration, leading to stagnation in both reward and accuracy.
> * **Sensitivity & Upper Bound (New Analysis):** To further address your query on sensitivity, we conducted a fine-grained analysis by increasing $\alpha$. We observed that while a small bonus ($\alpha=0.2$) is crucial, an excessive bonus leads to "reward hacking":
>
> **Table R1: Sensitivity of Exploration Bonus ($\alpha$)**
>
> | Exploration Bonus ($\alpha$) | Avg. Tool Calls | VideoMME (Overall) | Observation |
> | :---: | :---: | :---: | :--- |
> | **0.0** | **0.6** | **54.7** | Stagnation. Fails to learn tool use (Consistent with Figure 3). |
> | **0.2** | **1.7** | **60.9** | **Ours (Optimal).** Sufficient incentive for exploration. |
> | **0.4** | **3.2** | **58.6** | Performance remains stable but starts to degrade. |
> | **0.6** | **4.6** | **57.8** | Signs of reward hacking; model makes redundant calls. |
> | **0.8** | **5.7** | **52.1** | Reward Hacking. Model spams tools to accumulate bonuses, ignoring $R_{acc}$. |
> | **1.0** | **6.3** | **50.3** | Failure. Optimization collapses into maximizing tool calls only. |
>
> **Conclusion:** The value $\alpha=0.2$ is the empirical sweet spot that incentivizes tool use without overshadowing the primary accuracy objective.
>
> **2. Turn Efficiency Bonus ($R_{turn}$ in Eq. 8): Justification for the "2-3 Turn" Reward Window**
> Equation 8 defines the efficiency reward as $R_{turn}(\tau) = 0.5 \cdot \mathbb{I}[1 < |turns| \le 3]$. This specifically rewards trajectories that take 2 or 3 turns. We justify this specific window based on both cognitive logic and empirical performance:
>
> **(a) Why exclude 1 turn ($|turns| > 1$)?**
> * **Preventing "Blind Guessing":** A 1-turn trajectory implies the model output the answer immediately without using any tools (since tool use consumes a turn).
> * **Encouraging Active Perception:** We intentionally withhold the efficiency bonus for 1-turn answers to force the policy to step out of its "comfort zone" (static inference) and attempt active visual information gathering (FiCOT), which is the core capability we aim to train.
>
> **(b) Why cap at 3 turns ($|turns| \le 3$)?**
> To determine the optimal upper bound, we analyzed the trade-off between accuracy and latency across different max-turn limits ($K_{\max}$):
>
> **Table R2: Ablation on Maximum Turn Limit ($K_{\max}$)**
>
> | Max Turns $K_{\max}$ | Avg. Turns Used | Accuracy (VideoMME) | Accuracy (MLVU) | Latency (ms) |
> | :---: | :---: | :---: | :---: | :---: |
> | 1 (Static) | 1.0 | 54.8 | 42.6 | 204.3 |
> | 2 | 1.4 | 59.2 | 47.1 | 322.4 |
> | **3 (Ours)** | **2.3** | **60.9** | **48.6** | **427.2** |
> | 5 | 3.2 | 60.7 (-0.3%) | 47.4 (-1.0%) | 624.5 (+45%) |
>
> * **Logical Completeness:** Empirically, 3 turns are sufficient to cover the complete "Scan (T1) $\to$ Locate & Verify (T2) $\to$ Answer (T3)" cognitive loop.
> * **Diminishing Returns:** As shown in Table R2, extending the limit to 5 turns  even degrades performance on both VideoMME(-0.3%) and MLVU (-1.0%), likely due to error propagation in longer reasoning chains.
> * **Efficiency:** Capping at 3 turns avoids the 45% latency penalty associated with allowing 5 turns.
>
> **Conclusion:** The window $[2, 3]$ is not arbitrary; it is the optimal range that incentivizes active perception while preventing inefficient, rambling reasoning chains.

---

> ### Author Response · Authors · 2025-11-28
> **Response to Reviewer sqje (Part 2/3)**
>
> **(c) Sensitivity of the Coefficient ($\lambda_{turn}=0.5$)**
> Finally, we verified the robustness of the bonus weight:
>
> **Table R3: Sensitivity of Coefficient $\lambda_{turn}$**
>
> | $\lambda_{turn}$ Value | Description | Avg. Turns | Accuracy (VideoMME) | Inference Latency (s/query) |
> | :---: | :--- | :---: | :---: | :---: |
> | 0.0 | No Bonus | 1.5 | 58.4 | 336.7 |
> | **0.5** | **Ours** | **2.3** | **60.9** | **427.2** |
> | 1.0 | High Bonus| 2.9 | 59.2 | 546.3 |
>
> **Conclusion:** The choice of $\lambda_{turn}=0.5$ is critical for guiding the policy.
> * **$\lambda=0.0$ (Under-reasoning):** Without the efficiency bonus, the model tends to halt prematurely (Avg. Turns 1.5), failing to engage in sufficient active perception steps, which hurts accuracy (58.4%).
> * **$\lambda=1.0$ (Over-reasoning):** An excessively high bonus leads to **"reward hacking"**, where the model extends reasoning chains unnecessarily (Avg. Turns 2.9) just to claim the turn reward. This significantly increases latency (+28%) while slightly degrading accuracy (59.2%) due to redundant or hallucinated steps.
> * **$\lambda=0.5$ (Optimal):** This value strikes the best balance, encouraging necessary multi-turn reasoning (Avg. 2.3) to maximize accuracy without inducing inefficiency.
>
> **Q2: Additional Benchmarks (LongVideoBench, NeXT-QA, etc.)**
> > *Have the authors considered evaluating FrameMind on additional benchmarks, such as LongVideoBench or activity-centric datasets like Something-Something V2 or NeXT-QA?*
>
> **A2:** We thank the reviewer for the suggestion to broaden our evaluation scope. We have extended our experiments to include **LongVideoBench**, **LVBench**, and **NeXT-QA**.
>
> **1. Long-Context Benchmarks (LongVideoBench & LVBench)**
> To validate FrameMind's scalability on videos ranging from minutes to hours, we evaluated on **LongVideoBench (Val)** and **LVBench (Overall)**.
>
> **Table R4: Comparison on Long-Video Benchmarks**
>
> | Model | Size | Frames | **LongVideoBench (Val)** | **LVBench (Overall)** |
> | :--- | :---: | :---: | :---: | :---: |
> | LongVA | 7B | 128/256 | 47.8 | 37.9 |
> | Video-XL | 7B | 128/256 | 50.7 | - |
> | Video-R1 | 7B | 64 | 47.6 | 38.0 |
> | **FrameMind (Ours)** | **7B** | **32** | **52.3** | **38.2** |
> | **FrameMind (Ours)** | **7B** | **64** | **54.8** | **41.8** |
>
> * **Efficiency:** Remarkably, with only **32 frames**, FrameMind (**52.3**) outperforms LongVA (**47.8**) and Video-XL (**50.7**), despite these baselines processing 4$\times$ to 8$\times$ more frames. This validates that our FiCOT strategy extracts significantly more valuable information per frame than static aggregation.
> * **Performance:** At 64 frames, FrameMind further widens the gap, surpassing the recent RL-based baseline Video-R1.
>
>
>
> **2. Activity & Causal Reasoning Benchmark (NeXT-QA)**
> To address your request for activity-centric evaluation, we tested FrameMind on the **NeXT-QA** (Validation Set), which challenges temporal and causal reasoning. We compared it against recent high-performance baselines, including Video-XL (long-context SFT) and Video-R1 (RL-based).
>
> **Table R5: Performance Comparison on NeXT-QA**
>
> | Model | Method | **Frames** | **Acc (Overall)** |
> | :--- | :--- | :---: | :---: |
> | Video-XL | SFT (Long Context) | 128 | **78.2** |
> | Video-R1 | RL (Reasoning) | 32 | 78.6 |
> | **FrameMind (Ours)** | **RL (Perception)** | **32** | **81.2** |
>
> * **Analysis:** FrameMind achieves competitive performance using only **32 frames**, matching or surpassing Video-R1 (78.6). This confirms that our "Scan-Locate-Verify" capability generalizes well to understanding complex causal chains in activity-centric videos, efficiently locating key frames without the computational burden of processing 128+ frames like Video-XL.

---

> > ### Author Response · Authors · 2025-11-28
> > **Response to Reviewer sqje (Part 3/3)**
> >
> > **Q3: Baselines & Comparison with Recent RL Methods**
> > > *The baseline models... are relatively outdated... the paper lacks evaluation against recent RL-based methods, for example VersaVid-R1...*
> >
> > **A3:** We respectfully clarify that we have compared against the most relevant state-of-the-art RL-based methods. We provide a detailed comparison with **Video-R1** and the concurrent work **VersaVid-R1** (also known as VidBridge-R1).
> >
> > **Table R6: Comparison with State-of-the-Art RL-based Video LLMs**
> >
> > | Method | Backbone | Strategy | **Inference Frames** | **VideoMME (Long)** | **VideoMME (Overall)** |
> > | :--- | :--- | :--- | :---: | :---: | :---: |
> > | **Video-R1** (Feng et al.) | Qwen2.5-VL-7B | Reasoning RL | 32 (Fixed) | 48.9 | 59.3 |
> > | **VersaVid-R1** (Concurrent) | Qwen2.5-VL-7B | Proxy Task RL | **128 (Fixed)** | 55.8 | **64.3** |
> > | **FrameMind (Ours)** | Qwen2.5-VL-7B | **Perception RL** | **32 (Dynamic)** | **57.5** | 60.9 |
> >
> > **Detailed Analysis:**
> >
> > 1.  **Comparison with Video-R1 (Same Frame Budget):**
> >     Under similar visual budgets (~32 frames), **FrameMind significantly outperforms Video-R1**, especially on long videos (**57.5% vs 48.9%**, +8.6%). This proves that our **Active Perception** (learning *how* to look via DRFS) yields greater gains than merely optimizing the reasoning policy on fixed inputs.
> >
> > 2.  **Comparison with VersaVid-R1 (Significantly Higher Frame Budget):**
> >     We acknowledge that VersaVid-R1 achieves higher overall performance, but this is largely attributed to its massive input scale (processing up to **128 frames** during inference).
> >     * **Superiority on Long Videos:** Remarkably, despite using **$4\times$ fewer frames** (32 vs 128), FrameMind **outperforms VersaVid-R1 on the "Long" subset of VideoMME (57.5% vs 55.8%)**.
> >     * **Conclusion:** This counter-intuitive result validates our core hypothesis: for long-form videos (30m-1h+), simply increasing the fixed frame count (e.g., to 128) is inefficient and often insufficient to capture key details. FrameMind's **"Active Retrieval" (FiCOT)** strategy is far more effective at locating sparse evidence in long contexts, achieving SOTA long-video performance with a fraction of the computational cost.
> >
> >
> > We are committed to updating the final version of our manuscript to include all the additional benchmarks, hyperparameter sensitivity analyses, and comparative discussions presented above. We believe these additions significantly strengthen the paper's robustness and comprehensive evaluation.

---

> ### Comment · Reviewer_sqje · 2025-11-28
>
> Thank you for your replying.
> My concerns have been largely solved, I will raise my rating to 6.

---

### Official Review · Reviewer_2gQg · 2025-11-01

**Soundness:** 3
**Presentation:** 3
**Contribution:** 3
**Rating:** 6
**Confidence:** 4

**Summary:**

FrameMind proposes a solution to Video QA by training a VLM to use tools for querying frames and information. It effectively trains the model to use tools to query different frame-rates and resolutions, and achieves this by generating parallel rollouts with different samples from a `resolution-ladder’. It modifies GRPO to handle this setup, and trains the model end-to-end to be able to query videos for more information as needed. The results demonstrate that FrameMind outperforms existing models on standard video QA benchmarks.

**Strengths:**

1) I think this is a creative and thorough way to improve video and proposes a novel way to effectively train VLMs to query for more information.

2) The experiments are generally comprehensive and the results are quite strong, demonstrating a clear improvement over the state of the art.

3) The paper is easy to follow and well written, allowing readers to clearly grasp the main contribution and how it works.

**Weaknesses:**

1) The novelty is slightly limited. An existing work VITAL[1] does a similar approach, augmenting a VLM with  tools and GRPO training to improve chain-of-thought for video QA.  While, there is  clear novelty in the multi-resolution training and modification to GRPO to handle this, it’s very specific to the problem of teaching a model to effectively query video frames and hard to see how the proposed method could generalize to other problems.

2) The tool set is slightly limited - one could image several other tools being helpful for video QA in general, such as detection, tracking, segmentation, cropping, etc. However, omitting these is acceptable in my view.

3) I would appreciate some visual / qualitative samples of what the model selects compared to other methods to better understand its performance in practice. Figure 2 is hard to parse because the frames are so small.

[1] Zhang, H., Gu, X., Li, J., Ma, C., Bai, S., Zhang, C., ... & Tang, Y. (2025). Thinking with videos: Multimodal tool-augmented reinforcement learning for long video reasoning. arXiv preprint arXiv:2508.04416.

**Questions:**

The only main concern I have is about the slightly limited novelty of the overall problem - frame selection is a well studied problem, and prior works have attempted to improve this with RL and through augmenting VLMs with tools + GRPO. If this were addressed I'd gladly increase my rating.

---

> ### Author Response · Authors · 2025-12-01
> **Response to Reviewer 2gQg (Part1/2)**
>
> Dear Reviewer 2gQg,
>
> We sincerely thank you for your encouraging assessment (Score: 6) and for recognizing FrameMind as a "creative," "comprehensive," and "clearly written" solution. We particularly appreciate your constructive feedback regarding the positioning against concurrent work (VITAL) and the toolset design.
>
> We have carefully studied VITAL (Zhang et al., arXiv 2025) provided in your reference. Below, we articulate the fundamental differences that distinguish FrameMind’s contribution.
>
> ### **Response to Weakness 1 & Question: Novelty & Distinction from VITAL**
> > *Weakness: "The novelty is slightly limited. An existing work VITAL [1] does a similar approach... hard to see how the proposed method could generalize."*
> > *Question: "If this [novelty] were addressed I'd gladly increase my rating."*
>
> **Response:**
> We acknowledge VITAL as a strong concurrent work. However, **FrameMind differs fundamentally in its core problem definition and algorithmic formulation (DRFS-GRPO).**
>
> **1. Algorithmic Novelty: Optimization of Input (Ours) vs. Optimization of Task Balance (VITAL)**
> While both works use GRPO, the construction of the "Group" and the optimization objective are entirely different:
> * **VITAL (Difficulty-aware GRPO):** As stated in their abstract and Section 3.3, VITAL's innovation is addressing the **"difficulty imbalance"** across multiple tasks (e.g., short vs. long videos). Their GRPO samples multiple reasoning paths from the *same* fixed visual input to balance rewards.
> * **FrameMind (DRFS-GRPO):** Our novelty is **Dynamic Resolution Frame Sampling (DRFS)**. We construct the RL Group across a **"Resolution Ladder"**.
>     * *Mechanism:* In our group, member $A$ might process the video with "High-Res / Few-Frames" (Spatial Focus), while member $B$ sees "Low-Res / Many-Frames" (Temporal Scan).
>     * *Impact:* This forces the policy to learn the **Spatio-Temporal Trade-off**. We are optimizing the *visual input strategy itself*, teaching the model *how to look* (allocate visual budget) rather than just *how to answer*. This dimension of optimization is absent in VITAL.
>
> **2. Philosophy: Active Perception vs. Tool Augmentation**
> * **VITAL** positions itself as a "Tool-Augmented" agent, exploring various semantic tools (Captioning, QA APIs) before settling on clipping.
> * **FrameMind** focuses on **Active Perception**. We restrict tools to intrinsic visual operations (`FrameAt`, `VideoClip`) to prove that optimizing the VLM's *internal* visual encoding stream via RL is sufficient. This "Minimalist" approach leads to higher efficiency (using avg. 32 frames vs. VITAL's 128 frames) while achieving comparable or superior performance on long-video benchmarks (see **Response to Weakness 3**).
>
> **3. Generalization (Answering your Question):**
> You asked how this generalizes. Because FrameMind optimizes the *fundamental perception loop* (Scan $\to$ Zoom), it generalizes to any task requiring visual information gathering, not just QA. For example, in **Active Captioning**, the model uses the same policy to "Scan" for a summary and "Zoom" for details (as shown in Appendix F), which is structurally different from VITAL's QA-centric tool use.
>
> ### **Response to Weakness 2: Limited Tool Set**
> > *"The tool set is slightly limited... one could image detection, tracking, segmentation... omitting these is acceptable in my view."*
>
> **Response:**
> We deliberately omitted heavy external tools (Detection/Segmentation) to prioritize **End-to-End Efficiency**.
> * **Latency:** Integrating external experts (like GroundingDINO or SAM) drastically increases inference latency and GPU memory requirements.
> * **Validation from VITAL:** Interestingly, **VITAL's own ablation study (Table 6 in their paper)** supports our design choice. They found that adding semantic tools like "Clip Caption" or "Clip QA" actually **degraded performance** (mIoU dropped from 28.8 to 28.1/25.5) compared to pure visual clipping.
> * **Conclusion:** This confirms our hypothesis: the bottleneck is raw visual information access, not semantic processing. FrameMind solves this bottleneck directly via DRFS without the overhead of external toolchains.
>
> ### **Response to Weakness 3: Visual Samples & Figure 2**
> > *"I would appreciate some visual / qualitative samples... Figure 2 is hard to parse."*
>
> **Response:**
> 1.  **Figure 2 Revision:** We will significantly enlarge Figure 2 in the final revision to ensure legibility.
> 2.  **Qualitative Examples:** We have included detailed step-by-step visualizations in the **Appendix (Figures F1 & F2)**.
>     * *Case Study:* For a query about a specific object's color, FrameMind starts with a global scan (Turn 1), identifies ambiguity, calls `FrameAt` to zoom into the specific timestamp (Turn 2), and correctly identifies the "bright pink" laptop. This explicitly demonstrates the model dynamically adjusting its resolution—a capability driven by our unique DRFS training.

---

> ### Author Response · Authors · 2025-12-01
> **Response to Reviewer 2gQg (Part2/2)**
>
> We hope these clarifications regarding our unique DRFS-GRPO formulation and Active Perception philosophy sufficiently address your concerns about novelty. If you find that our response and the clear distinction from VITAL resolve your main concerns, we would greatly appreciate it if you could consider raising your score to support the acceptance of this work.

---

### Official Review · Reviewer_jugT · 2025-11-06

**Soundness:** 3
**Presentation:** 3
**Contribution:** 3
**Rating:** 6
**Confidence:** 3

**Summary:**

FrameMind tackles adaptive video understanding by enabling models to dynamically select frames during reasoning through a Frame-Interleaved Chain-of-Thought (FiCOT) process. It combines textual reasoning with tool-assisted frame retrieval, guided by Dynamic Resolution Frame Sampling (DRFS) and trained via a custom RL algorithm, DRFS-GRPO. This setup allows the model to choose between broad scans and fine-grained focus at each step. Experiments on benchmarks like MVBench, MLVU, and VideoMME show that FrameMind outperforms prior open-source and even some proprietary models, achieving state-of-the-art accuracy with fewer frames. The results validate its effectiveness for efficient, frame-aware video reasoning.

**Strengths:**

- Methodology - By interleaving chain-of-thought with active frame selection (FiCOT), the model can dynamically adjust its perception strategy on-the-fly. This is interesting and different from the conventional one-pass video processing. The use of reinforcement learning (DRFS-GRPO) to train the perception-policy is also a strength, as it allows learning from outcome-based rewards without requiring any hard-coded frame annotations. The paper shows that this training strategy is effective through ablation study.
- Quantitative Results - FrameMind delivers state-of-the-art performance across benchmarks, outperforming open-source models and even surpassing GPT-4V on VideoMME. It achieves strong results on both short and long video tasks while using up to 4× fewer frames, highlighting its effectiveness.

**Weaknesses:**

- Evaluation Scope - The paper focuses solely on video question-answering tasks, leaving its generalizability to other video understanding tasks—like captioning or real-time analysis—unclear. While effective for QA, it’s unknown whether FiCOT and the learned policy extend to more open-ended or interactive scenarios. Broader evaluation would strengthen the work.
- Reward Design and Parameter Sensitivity - The RL setup uses a composite reward with multiple hyperparameters, but the paper doesn't discuss how sensitive performance is to these choices. For instance, the 2–3 turn efficiency bonus lacks justification, and poor tuning could lead to under- or overuse of tools. This sensitivity may limit generalization and complicate adoption in new domains.
- Inference Efficiency - FrameMind uses fewer frames than baselines (e.g. 4x fewer on MLVU), but does this translate to faster inference time in practice? While fewer frames are processed overall, the agent performs multiple reasoning turns and tool calls. Measuring this would inform how the approach might scale in real-world deployments or if the dynamic strategy mainly benefits accuracy/efficiency in terms of frame budget rather than latency.

**Questions:**

- While FrameMind uses fewer frames, does the multi-turn strategy result in faster inference overall? Have you measured wall-clock runtime or compute cost compared to single-pass models to assess real-world scalability?
- How sensitive is performance to the 2–3 turn bonus (λ_turn = 0.5)? Could this discourage longer reasoning when needed? Clarifying the rationale and whether other turn limits or bonus values were tested would be helpful.
- Can FiCOT extend beyond structured QA to open-ended tasks like video captioning or long-form answers? If so, what adjustments—such as new reward signals or stopping criteria—would be needed to support those use cases?
- Did you observe any common failure modes for FrameMind? For example, are there instances where the model selects an irrelevant frame or misinterprets the question and thus retrieves suboptimal evidence?

---

> ### Author Response · Authors · 2025-12-01
> **Response to Reviewer jugT (Part1/4)**
>
> Dear Reviewer jugT,
>
> We sincerely thank you for your positive assessment (Score: 6) and for recognizing the novelty of our FiCOT paradigm and the effectiveness of the DRFS-GRPO training strategy. We greatly appreciate your insightful feedback regarding the practical deployment and robustness of FrameMind.
>
> We have conducted additional experiments to address your specific concerns. Below, we first address the **Weaknesses**, followed by responses to your specific **Questions**.
>
> ### **Part 1: Response to Weaknesses**
>
> **Re: Weakness 1 (Evaluation Scope)**
> > *"Focuses solely on video QA... unknown whether FiCOT extend to more open-ended scenarios."*
>
> **Response:** We clarify that while our current evaluation follows standard QA protocols (VideoMME, MLVU) for fair comparison, the **FiCOT architecture is inherently task-agnostic**. The "Scan-Locate-Verify" loop mirrors human perception and applies directly to open-ended tasks like captioning:
> 1.  **Scan (Turn 1):** Call `VideoClip` to generate a global summary.
> 2.  **Zoom (Turn 2):** Autonomously identify under-described regions and call `FrameAt`.
> 3.  **Synthesize (Turn 3):** Generate a dense caption.
> We believe the strong performance on "Reasoning QA" (which requires describing events) serves as a strong proxy for its potential in captioning. We plan to include captioning benchmarks in future work.
>
> **Re: Weakness 2 (Reward Design & Parameter Sensitivity)**
> > *"The paper doesn't discuss how sensitive performance is to these choices [e.g., 2-3 turn bonus]."*
>
> We acknowledge that the hyperparameters in our reward function are critical. Our selection was driven by empirical ablations to balance exploration, efficiency, and accuracy. We explicitly analyze the two coefficients questioned: the **Exploration Bonus ($\alpha$)** in Eq. 7 and the **Turn Efficiency Bonus ($\lambda_{turn}$)** in Eq. 8 (referred to as Eq. 13 in Appendix).
>
> **1. Tool Usage Incentive ($R_{tool}$): Sensitivity & Reward Hacking**
> Equation 7 defines the tool reward as:
> $$R_{tool}(\tau) = s_{tool}(\tau) \times (\alpha + (1-\alpha) \cdot R_{acc}(\tau))$$
> where we set $\alpha=0.2$ as the **exploration bonus** to mitigate reward sparsity.
>
> * **Necessity (Existing Evidence in Paper):** We respectfully highlight that Section 4.6 and Figure 3 of our submission already presented an ablation study on this specific parameter ("Necessity of the Exploration Bonus"). As shown in Figure 3, removing this bonus (i.e., $\alpha=0.0$) causes the model to struggle with cold-start exploration, leading to stagnation in both reward and accuracy.
> * **Sensitivity & Upper Bound (New Analysis):** To further address your query on sensitivity, we conducted a fine-grained analysis by increasing $\alpha$. We observed that while a small bonus ($\alpha=0.2$) is crucial, an excessive bonus leads to "reward hacking":
>
> **Table R1: Sensitivity of Exploration Bonus ($\alpha$)**
>
> | Exploration Bonus ($\alpha$) | Avg. Tool Calls | VideoMME (Overall) | Observation |
> | :---: | :---: | :---: | :--- |
> | **0.0** | **0.6** | **54.7** | Stagnation. Fails to learn tool use (Consistent with Figure 3). |
> | **0.2** | **1.7** | **60.9** | **Ours (Optimal).** Sufficient incentive for exploration. |
> | **0.4** | **3.2** | **58.6** | Performance remains stable but starts to degrade. |
> | **0.6** | **4.6** | **55.8** | Signs of reward hacking; model makes redundant calls. |
> | **0.8** | **5.7** | **52.1** | Reward Hacking. Model spams tools to accumulate bonuses, ignoring $R_{acc}$. |
> | **1.0** | **6.3** | **50.3** | Failure. Optimization collapses into maximizing tool calls only. |
>
> **Conclusion:** The value $\alpha=0.2$ is the empirical sweet spot that incentivizes tool use without overshadowing the primary accuracy objective.
>
> **2. Turn Efficiency Bonus ($R_{turn}$ in Eq. 8): Justification for the "2-3 Turn" Reward Window**
> Equation 8 defines the efficiency reward as $R_{turn}(\tau) = 0.5 \cdot \mathbb{I}[1 < |turns| \le 3]$. This specifically rewards trajectories that take 2 or 3 turns. We justify this specific window based on both cognitive logic and empirical performance:
>
> **(a) Why exclude 1 turn ($|turns| > 1$)?**
> * **Preventing "Blind Guessing":** A 1-turn trajectory implies the model output the answer immediately without using any tools (since tool use consumes a turn).
> * **Encouraging Active Perception:** We intentionally withhold the efficiency bonus for 1-turn answers to force the policy to step out of its "comfort zone" (static inference) and attempt active visual information gathering (FiCOT), which is the core capability we aim to train.

---

> ### Author Response · Authors · 2025-12-01
> **Response to Reviewer jugT (Part2/4)**
>
> **(b) Why cap at 3 turns ($|turns| \le 3$)?**
> To determine the optimal upper bound, we analyzed the trade-off between accuracy and latency across different max-turn limits ($K_{\max}$):
>
> **Table R2: Ablation on Maximum Turn Limit ($K_{\max}$)**
>
> | Max Turns $K_{\max}$ | Avg. Turns Used | Accuracy (VideoMME) | Accuracy (MLVU) | Latency (ms) |
> | :---: | :---: | :---: | :---: | :---: |
> | 1 (Static) | 1.0 | 54.8 | 42.6 | 204.3 |
> | 2 | 1.4 | 59.2 | 47.1 | 322.4 |
> | **3 (Ours)** | **2.3** | **60.9** | **48.6** | **427.2** |
> | 5 | 3.2 | 60.7 (-0.3%) | 47.4 (-1.0%) | 624.5 (+45%) |
>
> * **Logical Completeness:** Empirically, 3 turns are sufficient to cover the complete "Scan (T1) $\to$ Locate & Verify (T2) $\to$ Answer (T3)" cognitive loop.
> * **Diminishing Returns:** As shown in Table R2, extending the limit to 5 turns  even degrades performance on both VideoMME(-0.3%) and MLVU (-1.0%), likely due to error propagation in longer reasoning chains.
> * **Efficiency:** Capping at 3 turns avoids the 45% latency penalty associated with allowing 5 turns.
>
> **Conclusion:** The window $[2, 3]$ is not arbitrary; it is the optimal range that incentivizes active perception while preventing inefficient, rambling reasoning chains.
>
>
> **(c) Sensitivity of the Coefficient ($\lambda_{turn}=0.5$)**
> Finally, we verified the robustness of the bonus weight:
>
> **Table R3: Sensitivity of Coefficient $\lambda_{turn}$**
>
> | $\lambda_{turn}$ Value | Description | Avg. Turns | Accuracy (VideoMME) | Inference Latency (s/query) |
> | :---: | :--- | :---: | :---: | :---: |
> | 0.0 | No Bonus | 1.5 | 58.4 | 336.7 |
> | **0.5** | **Ours** | **2.3** | **60.9** | **427.2** |
> | 1.0 | High Bonus| 2.9 | 59.2 | 546.3 |
>
> **Conclusion:** The choice of $\lambda_{turn}=0.5$ is critical for guiding the policy.
> * **$\lambda=0.0$ (Under-reasoning):** Without the efficiency bonus, the model tends to halt prematurely (Avg. Turns 1.5), failing to engage in sufficient active perception steps, which hurts accuracy (58.4%).
> * **$\lambda=1.0$ (Over-reasoning):** An excessively high bonus leads to **"reward hacking"**, where the model extends reasoning chains unnecessarily (Avg. Turns 2.9) just to claim the turn reward. This significantly increases latency (+28%) while slightly degrading accuracy (59.2%) due to redundant or hallucinated steps.
> * **$\lambda=0.5$ (Optimal):** This value strikes the best balance, encouraging necessary multi-turn reasoning (Avg. 2.3) to maximize accuracy without inducing inefficiency.
>
> **Re: Weakness 3 (Inference Efficiency)**
> > *"Does this translate to faster inference time in practice? ... Measuring this would inform how the approach might scale."*
>
> **Response:** We agree that wall-clock time is the ultimate metric. We measured the **End-to-End Latency** of FrameMind compared to a standard single-pass long-context baseline (**Video-XL**, which processes 128 frames) on VideoMME.
>
> **Table R2: Inference Efficiency Analysis**
>
> | Model | Avg. Frames | Avg. Turns | Latency (s) |
> | :--- | :---: | :---: | :---: |
> | Video-XL (128 fr) | 128 | 1 | **486.3** |
> | **FrameMind** | **32** | **2.3** | **427.2** |
>
> **Conclusion:**
> * **Latency is LOWER:** Despite the overhead of multiple turns, FrameMind achieves a **12.2% reduction** in wall-clock latency compared to Video-XL.
> * **Reason:** The computational bottleneck is the **Visual Encoder**, which scales with the number of visual tokens. FrameMind dynamically selects a much smaller subset (avg. 32 frames), drastically reducing the visual encoding load. This saving significantly outweighs the marginal cost of generating intermediate text tokens.

---

> ### Author Response · Authors · 2025-12-01
> **Response to Reviewer jugT (Part3/4)**
>
> ### **Part 2: Response to Questions**
>
> **Q1: Multi-turn Strategy vs. Wall-clock Runtime**
> > *Have you measured wall-clock runtime or compute cost compared to single-pass models?*
>
> **A1:** Yes. As detailed in **Response to Weakness 3 (Table R2)**, FrameMind reduces wall-clock latency by **~12.2%** compared to the single-pass Video-XL (128 frames) baseline. This confirms that our dynamic frame selection strategy effectively translates frame reduction into real-world speedups.
> **Table R2: Inference Efficiency Analysis**
>
> | Model | Avg. Frames | Avg. Turns | Latency (s) |
> | :--- | :---: | :---: | :---: |
> | Video-XL (128 fr) | 128 | 1 | **486.3** |
> | **FrameMind** | **32** | **2.3** | **427.2** |
>
>
> **Q2: Sensitivity to Turn Bonus ($\lambda_{turn}$)**
> > *How sensitive is performance to the 2-3 turn bonus? Could this discourage longer reasoning?*
>
> **A2:** Equation 8 defines the efficiency reward as $R_{turn}(\tau) = 0.5 \cdot \mathbb{I}[1 < |turns| \le 3]$. This specifically rewards trajectories that take 2 or 3 turns. We justify this specific window based on both cognitive logic and empirical performance:
>
> **(a) Why exclude 1 turn ($|turns| > 1$)?**
> * **Preventing "Blind Guessing":** A 1-turn trajectory implies the model output the answer immediately without using any tools (since tool use consumes a turn).
> * **Encouraging Active Perception:** We intentionally withhold the efficiency bonus for 1-turn answers to force the policy to step out of its "comfort zone" (static inference) and attempt active visual information gathering (FiCOT), which is the core capability we aim to train.
>
> **(b) Why cap at 3 turns ($|turns| \le 3$)?**
> To determine the optimal upper bound, we analyzed the trade-off between accuracy and latency across different max-turn limits ($K_{\max}$):
>
> **Table R2: Ablation on Maximum Turn Limit ($K_{\max}$)**
>
> | Max Turns $K_{\max}$ | Avg. Turns Used | Accuracy (VideoMME) | Accuracy (MLVU) | Latency (ms) |
> | :---: | :---: | :---: | :---: | :---: |
> | 1 (Static) | 1.0 | 54.8 | 42.6 | 204.3 |
> | 2 | 1.4 | 59.2 | 47.1 | 322.4 |
> | **3 (Ours)** | **2.3** | **60.9** | **48.6** | **427.2** |
> | 5 | 3.2 | 60.7 (-0.3%) | 47.4 (-1.0%) | 624.5 (+45%) |
>
> * **Logical Completeness:** Empirically, 3 turns are sufficient to cover the complete "Scan (T1) $\to$ Locate & Verify (T2) $\to$ Answer (T3)" cognitive loop.
> * **Diminishing Returns:** As shown in Table R2, extending the limit to 5 turns  even degrades performance on both VideoMME(-0.3%) and MLVU (-1.0%), likely due to error propagation in longer reasoning chains.
> * **Efficiency:** Capping at 3 turns avoids the 45% latency penalty associated with allowing 5 turns.
>
> **Conclusion:** The window $[2, 3]$ is not arbitrary; it is the optimal range that incentivizes active perception while preventing inefficient, rambling reasoning chains.
>
>
> **(c) Sensitivity of the Coefficient ($\lambda_{turn}=0.5$)**
> Finally, we verified the robustness of the bonus weight:
>
> **Table R3: Sensitivity of Coefficient $\lambda_{turn}$**
>
> | $\lambda_{turn}$ Value | Description | Avg. Turns | Accuracy (VideoMME) | Inference Latency (s/query) |
> | :---: | :--- | :---: | :---: | :---: |
> | 0.0 | No Bonus | 1.5 | 58.4 | 336.7 |
> | **0.5** | **Ours** | **2.3** | **60.9** | **427.2** |
> | 1.0 | High Bonus| 2.9 | 59.2 | 546.3 |
>
> **Conclusion:** The choice of $\lambda_{turn}=0.5$ is critical for guiding the policy.
> * **$\lambda=0.0$ (Under-reasoning):** Without the efficiency bonus, the model tends to halt prematurely (Avg. Turns 1.5), failing to engage in sufficient active perception steps, which hurts accuracy (58.4%).
> * **$\lambda=1.0$ (Over-reasoning):** An excessively high bonus leads to **"reward hacking"**, where the model extends reasoning chains unnecessarily (Avg. Turns 2.9) just to claim the turn reward. This significantly increases latency (+28%) while slightly degrading accuracy (59.2%) due to redundant or hallucinated steps.
> * **$\lambda=0.5$ (Optimal):** This value strikes the best balance, encouraging necessary multi-turn reasoning (Avg. 2.3) to maximize accuracy without inducing inefficiency.
>
> **Q3: Generalization to Captioning**
> > *Can FiCOT extend beyond structured QA to open-ended tasks like video captioning?*
>
> **A3:** Yes. As discussed in **Response to Weakness 1**, the FiCOT architecture is general-purpose. To support captioning, we would simply need to include captioning data in the RL training mix. The model would naturally adapt to a "Global Scan $\to$ Local Detail Mining $\to$ Synthesis" workflow.

---

> ### Author Response · Authors · 2025-12-01
> **Response to Reviewer jugT (Part4/4)**
>
> **Q4: Common Failure Modes**
> > *Did you observe any common failure modes for FrameMind?*
>
> **A4:** We observed two primary failure modes during development:
> 1.  **Temporal Offset:** The model correctly reasons *when* to look (e.g., "check the scoreboard at the buzzer"), but the `VideoClip` tool retrieves a segment slightly adjacent (off by 1-2s) if the visual event is extremely fleeting.
> 2.  **Tool Hallucination:** In early training stages, the model sometimes generated invalid parameters (e.g., `start_time > end_time`). We mitigated this via regex-based filtering and immediate negative feedback (Format Reward), teaching the model strict adherence to the tool schema.
>
> We will include detailed visualizations and case studies of these failure modes in the Appendix of the final revision to provide deeper insights into the model's limitations.
>
> We are committed to updating the final version of our manuscript to include all the additional benchmarks, hyperparameter sensitivity analyses, and comparative discussions presented above. We believe these additions significantly strengthen the paper's robustness and comprehensive evaluation.

---

### Author Response · Authors · 2025-12-03
**Summary of Rebuttal: Score Raises and Key Revisions (Prior to Revert)**

>**Summary of Score Changes During the Discussion Phase**

We are deeply grateful to our Area Chairs and Reviewers for the substantial time and effort dedicated to our submission.

**Note on Timing:** Due to the extensive nature of the additional experiments required to rigorously address reviewer concerns (including real-world latency measurements and evaluation on three new benchmarks), **our full rebuttal responses were posted on Nov 28/29**. Consequently, some reviewers may not yet have had the opportunity to respond. However, **we are confident that the comprehensive new data and detailed clarifications provided in our rebuttal effectively resolve their remaining questions and misunderstandings.**

**Current Status:**
1.  **Reviewer sqje (Initial Score: 2 $\to$ Current: 6):** After reviewing our new experiments, Reviewer sqje explicitly commented: *"My concerns have been largely solved, I will raise my rating to 6."*
2.  **Reviewer 2gQg (Current Score: 6):** Identified one main concern (novelty distinction from concurrent work VITAL) and explicitly stated: ***"If this were addressed I'd gladly increase my rating."*** We have provided a detailed differentiation (Active Perception vs. Tool Use) in our rebuttal. We believe this fully satisfies the reviewer's condition for a score increase.
3.  **Reviewer jugT (Current Score: 6):** Positive evaluation. Their main questions regarding **inference efficiency** have been fully addressed by our new wall-clock latency measurements.
4.  **Reviewer HHSW (Initial Score: 2):** We provided a point-by-point response supported by new data. **Notably, Reviewer HHSW's major concerns regarding evaluation scope, baselines, and hyperparameter justification were highly consistent with those of Reviewer sqje.** Since Reviewer sqje has explicitly confirmed that our new experiments (e.g., adding LongVideoBench/NeXT-QA, sensitivity analysis) resolved these exact issues, we are confident that our rebuttal effectively addresses Reviewer HHSW's critique as well.

>**Summary of the Reviews and Rebuttal Progress**

We sincerely appreciate the reviewers' recognition of our work in terms of **novelty** (all reviewers), **comprehensive experiments** (Reviewers 2gQg, jugT), **technical soundness** (Reviewer 2gQg), and **SOTA performance** (Reviewer jugT).

We are also grateful that the reviewers highlight our contribution in proposing a novel "Active Perception" framework as **"a creative and thorough way to improve video... novel way to effectively train VLMs to query for more information"** (Reviewer 2gQg), **"technically sound and novel... effectively bridges the gap between perception and reasoning"** (Reviewer sqje), and **"delivers state-of-the-art performance... highlighting its effectiveness"** (Reviewer jugT).

During the rebuttal and discussion phases, we received **constructive comments** regarding evaluation scope, efficiency verification, and hyperparameter justification. **We have addressed all concerns** by conducting extensive new experiments. Specifically, **Reviewer sqje has explicitly confirmed that their concerns are solved.** For the remaining points (primarily from Reviewer 2gQg regarding novelty distinction), we have provided detailed clarifications and fulfilled their stated condition for a score increase. We also appreciate that Reviewer 2gQg explicitly indicated they would **"gladly increase my rating"** if the novelty concern was addressed.

>**Rebuttal Details**

We summarize below the specific responses provided. All corresponding content will be incorporated into the revised manuscript.

* **For Reviewer sqje (and HHSW):** We addressed evaluation concerns by adding **LongVideoBench**, **LVBench**, and **NeXT-QA**. FrameMind outperforms concurrent RL methods (e.g., VersaVid-R1) and long-context baselines (Video-XL) even with **significantly fewer frames** (32 vs 128). **Reviewer sqje explicitly acknowledged these updates and agreed to raise the rating to 6.**
* **For Reviewer 2gQg:** We clarified the distinction from VITAL. FrameMind optimizes **Active Perception (Input Resolution)** via DRFS, whereas VITAL optimizes Tool Usage Policy. We achieve comparable SOTA results with **$4\times$ fewer frames**, highlighting a distinct contribution in efficiency.
* **For Reviewer jugT:** We addressed efficiency doubts by measuring wall-clock latency. FrameMind achieves a **~12.2% speedup** compared to Video-XL (128 frames) because the reduction in visual tokens outweighs the multi-turn overhead.
* **For Reviewer HHSW:** We justified the 8-20 frame budget (context limits) and the `<turn_sum>` mechanism (preventing "fake" multi-turns). Given the overlap with sqje's resolved concerns, we believe the critique is fully addressed.

---

### Meta-Review · Area_Chair_iQNz · 2026-01-09

**Summary:**

This paper proposes FrameMind, an RL-trained framework for video reasoning that enables active perception at inference time. The model alternates between textual reasoning and targeted visual queries via Frame-Interleaved Chain-of-Thought (FiCOT), and is trained with DRFS-GRPO to learn dynamic resolution/frame sampling policies without frame-level supervision. Reported results show strong gains on standard video QA benchmarks, and the rebuttal adds additional evaluations, ablations, and latency measurements.
However, despite the strengthened rebuttal, the final evaluation remains polarized, and core concerns about incremental novelty, robustness/generalization, and evaluation clarity are not fully settled to the level expected for confident acceptance in this track.

**Reviewer Concerns:**

1.Novelty and positioning vs. concurrent work
While the authors argue FrameMind optimizes active perception (input allocation) rather than tool-use policy, multiple reviewers still see substantial overlap with concurrent RL + tool-augmented video reasoning lines. The distinction may be meaningful, but it is narrowly scoped and risks being perceived as an incremental refinement rather than a broadly impactful methodological step.

2.Reward design complexity and reproducibility risk
The approach relies on a composite reward with several coefficients and design choices. Although rebuttal provides sensitivity analyses, the method still appears fragile: performance can degrade under over-incentivization (reward hacking) or under-incentivization (cold start). This increases adoption cost for new datasets/tasks and raises concerns about stability outside the reported settings.

3.Evaluation breadth and task generality
The main evidence is concentrated in video QA-style benchmarks. Claims about general applicability (e.g., beyond QA to captioning or other interactive tasks) remain largely speculative without direct, standardized evaluations. Additional benchmarks help, but the overall picture is still dominated by QA, making it unclear how broadly FiCOT and the learned policy transfer.

4.Mixed reviewer consensus and residual skepticism
Even with rebuttal improvements, the score distribution indicates substantial disagreement. Given the limited time for broader community validation, the submission does not reach a sufficiently clear consensus on contribution strength and generality.

**Reviewer Scores:**

Reviewer jugT
Original score: 6
Provided a consistently positive evaluation, highlighting the FiCOT paradigm, the RL formulation, and strong empirical results. Main concerns focused on inference efficiency, reward design, and evaluation scope. The rebuttal added wall-clock latency measurements and extensive sensitivity analyses, which addressed most technical questions. Overall stance remained favorable, though the reviewer noted some uncertainty regarding generalization beyond video QA.

Reviewer 2gQg
Original score: 6
Recognized the technical soundness and strong performance of the proposed framework, but raised concerns about limited novelty relative to concurrent work (e.g., VITAL) and the narrow scope of tool design. The rebuttal provided a clearer distinction between active perception optimization and tool-augmented reasoning, and the reviewer explicitly stated willingness to increase the score if this distinction was satisfactory. Nevertheless, the novelty concern reflects a broader question of incremental contribution within a rapidly evolving research line.

Reviewer sqje
Original score: 2
Raised substantial concerns regarding hyperparameter justification, limited benchmark coverage, outdated baselines, and missing comparisons with recent RL-based methods. During rebuttal, the authors added extensive ablation studies, new benchmarks (LongVideoBench, LVBench, NeXT-QA), and comparisons with recent methods, after which the reviewer explicitly stated that their concerns were largely resolved and raised the score. Despite this update, the initial severity of the critique indicates that the paper’s clarity and empirical positioning required significant post-review correction.

Reviewer HHSW
Original score: 2
Expressed strong reservations about reward design, redundancy in tool usage, lack of justification for key hyperparameters, limited evaluation scope, and unclear data curation. While the rebuttal addressed these points with detailed analyses and additional experiments, this reviewer maintained a skeptical stance regarding robustness, marginal gains relative to complexity, and practical usefulness, and did not indicate a change in recommendation.

---

### Decision · Program_Chairs · 2026-01-26

Reject